# LRIG proteins regulate lipid metabolism via BMP signaling and affect the risk of type 2 diabetes

Carl Herdenberg[1], Pascal M. Mutie[2], Ola Billing [3], Ahmad Abdullah [1], Rona J. Strawbridge [4,5,6], Ingrid Dahlman[7], Simon Tuck[8], Camilla Holmlund[1], Peter Arner [7], Roger Henriksson[1], Paul W. Franks [2] & Håkan Hedman [1✉]

Leucine-rich repeats and immunoglobulin-like domains (LRIG) proteins have been implicated as regulators of growth factor signaling; however, the possible redundancy among mammalian LRIG1, LRIG2, and LRIG3 has hindered detailed elucidation of their physiological functions. Here, we show that *Lrig*-null mouse embryonic fibroblasts (MEFs) are deficient in adipogenesis and bone morphogenetic protein (BMP) signaling. In contrast, transforming growth factor-beta (TGF-β) and receptor tyrosine kinase (RTK) signaling appeared unaltered in *Lrig*-null cells. The BMP signaling defect was rescued by ectopic expression of LRIG1 or LRIG3 but not by expression of LRIG2. *Caenorhabditis elegans* with mutant *LRIG/sma-10* variants also exhibited a lipid storage defect. Human *LRIG1* variants were strongly associated with increased body mass index (BMI) yet protected against type 2 diabetes; these effects were likely mediated by altered adipocyte morphology. These results demonstrate that LRIG proteins function as evolutionarily conserved regulators of lipid metabolism and BMP signaling and have implications for human disease.

[1] Department of Radiation Sciences, Oncology, Umeå University, SE-90187 Umeå, Sweden. [2] Genetic and Molecular Epidemiology Unit, Department of Clinical Sciences, Lund University, Skane University Hospital, Malmo, Sweden. [3] Department of Surgical and Perioperative Sciences, Surgery, Umeå University, SE-90187 Umeå, Sweden. [4] Institute of Health and Wellbeing, University of Glasgow, Glasgow, UK. [5] Department of Medicine Solna, Karolinska Institutet, Stockholm, Sweden. [6] Health Data Research UK, University of Glasgow, Glasgow, UK. [7] Department of Medicine Huddinge, Karolinska Institutet, Stockholm, Sweden. [8] Umeå Center for Molecular Medicine, Umeå University, SE-90187 Umeå, Sweden. ✉email: hakan.hedman@umu.se

The mammalian leucine-rich repeats and immunoglobulin-like domains (LRIG/Lrig) protein family consists of three transmembrane proteins, LRIG1, LRIG2, and LRIG3[1]. The three mammalian LRIG paralogs appear to have both distinct and redundant functions, which is evident during mouse development[2]. Because *Lrig*-null (i.e., *Lrig1-/-;Lrig2-/-;Lrig3-/-*) mice are not viable, molecular investigations regarding the functions of the mammalian LRIG proteins have been hampered. Nevertheless, numerous reports have indicated that LRIG proteins are important etiological and prognostic factors in cancer[3,4]. In most cases, these roles have been attributed to the ability of LRIG1 to negatively regulate various receptor tyrosine kinases (RTKs)[5,6,7,8,9,10,11]. However, in the nematode *Caenorhabditis elegans* (*C. elegans*), the sole LRIG homolog, SMA-10, regulates body size by promoting bone morphogenetic protein (BMP) signaling[12,13]. Whether the mammalian LRIG proteins also regulate BMP signaling remains to be investigated.

The BMP signaling system is evolutionarily conserved and regulates major developmental and homeostatic processes[14-16]. The BMP families of cytokines and their receptors belong to the transforming growth factor-beta (TGF-β) and TGF-β receptor superfamilies, respectively. The human genome encodes at least 20 BMP ligands, three BMP type 1 receptors, and three type 2 receptors[15,17-19]. Upon ligand binding, the constitutively active type 2 receptor phosphorylates and activates an associated type 1 receptor, which, in turn, phosphorylates the downstream signaling mediators SMAD1, SMAD5, and SMAD8[15]. The phosphorylated SMAD1/5/8 complex then recruits the co-SMAD SMAD4 and is translocated into the nucleus, where it regulates expression of BMP-responsive genes[20]. In addition to the SMAD-mediated signaling pathway, BMPs may also initiate SMAD-independent signaling, including the activation of the MAP kinases ERK1/2, p38, and JNK[21,22]. Furthermore, BMP signaling is fine-tuned by regulatory proteins that either enhance or suppress signaling[15,20,23]. Although the BMP system has been extensively studied for decades, novel regulators and key signaling proteins may still await discovery.

Obesity constitutes a global epidemic and is a major risk factor for several conditions, including insulin resistance, type 2 diabetes, heart disease, and several forms of cancer[24-26]. Adipose tissue serves as a key regulator of energy homeostasis in humans[27]. Adipose tissue can expand in volume either by enlarging adipocyte size (hypertrophy) or by adipocyte proliferation (hyperplasia). Of these two processes, adipocyte hypertrophy is associated with an unfavorable metabolic profile, whereas hyperplasia may improve metabolic homeostasis due to the increased number of insulin-sensitive cells[27,28]. Adipocyte differentiation involves the sequential commitment of mesenchymal stem cells to preadipocytes followed by their numerical expansion and terminal differentiation into adipocytes[29,30]. In this process, BMP signaling is involved in the commitment of mesenchymal stem cells into preadipocytes[30,31], as well as in the choice between white or brown/beige adipocyte differentiation and adipocyte size[32]. *C. elegans*, on the other hand, lacks dedicated adipocytes;[33] however, evidence suggests that BMP signaling may also regulate lipid accumulation in the lipid-storing intestinal cells of *C. elegans*[34,35]. In mice, *Lrig3*-deficient animals display altered plasma lipid levels[36]. However, a direct link between LRIG or SMA-10 proteins and adipogenesis, lipid metabolism, or type 2 diabetes has not yet been studied.

In the present study, we generated *Lrig*-null mouse embryonic fibroblasts (MEFs) to analyze the physiological and molecular functions of LRIG proteins in isogenic cells, without the possibly confounding expression of endogenous LRIG proteins. By exploiting these cells, we demonstrated that mammalian LRIG proteins regulate adipogenesis and sensitize cells to low concentrations of BMPs. We also analyzed the *sma-10/LRIG* mutant *C. elegans* and showed that LRIG also regulates fat accumulation in the worm. Finally, we investigated possible associations between *LRIG1* single nucleotide polymorphisms (SNPs) and human metabolic traits and revealed a striking discordant association between common *LRIG1* variants, a reduced risk of type 2 diabetes, and an increased body mass index (BMI), which we showed was likely mediated by adipocyte morphology.

## Results

**Generation and characterization of *Lrig*-null MEFs.** To investigate the molecular functions of Lrig proteins, we generated *Lrig*-null cell lines by immortalizing MEFs that carried floxed *Lrig1*, *Lrig2*, and *Lrig3* alleles, followed by gene ablation via cell transduction with Cre recombinase-expressing adenoviruses. Thereby, we created four *Lrig*-null (herein also called *Lrig* triple knockout; TKO) MEF lines (TKO1-4) together with four corresponding wild-type control MEF lines (WT1-4). The stable ablation of *Lrig1*, *Lrig2*, and *Lrig3* was confirmed by polymerase chain reaction (PCR) genotyping (Supplementary Fig. 1a), Western blotting (Supplementary Fig. 1b), and a sensitive duplex droplet digital PCR (ddPCR) assay (Supplementary Table 1). With the ddPCR assay, the fraction of contaminating wild-type MEFs in the *Lrig*-null populations was analyzed by quantifying the relative gene copy numbers of the targeted *Lrig3* exon 1 versus a reference locus. This analysis showed that, in all the analyzed TKO MEF populations, more than 99.6% of the MEFs were *Lrig3*-negative (Supplementary Table 1). Importantly, long-term culturing of the TKO MEF lines for 60 days did not enrich for contaminating wild-type MEFs as assessed with any of the three genotyping methods (Supplementary Fig. 1; Supplementary Table 1). We then compared wild-type and *Lrig*-null MEF lines with regard to their proliferation and migration rates, morphology, and basic metabolic functions. The proliferation rates were similar between the wild-type and *Lrig*-null MEFs, both under standard cell culture conditions in 10% fetal bovine serum (FBS) (Supplementary Fig. 1c) and under proliferation-limiting FBS concentrations, although the *Lrig*-null MEFs showed a higher apparent proliferation rate than the wild-type MEFs specifically with 5% FBS (Supplementary Fig. 1d). The migratory rates of wild-type and *Lrig*-null MEFs were also similar, in both 10% FBS and in 0% FBS (Supplementary Fig. 1e). We were also unable to detect any apparent difference in cell morphology between the wild-type and *Lrig*-null MEF lines by light microscopy. Accordingly, flow cytometry analysis did not reveal any significant differences in forward or side scatter profiles between the wild-type and *Lrig*-null MEF lines (Supplementary Table 2). We then analyzed basic metabolic functions on a Seahorse XF analysis platform. These analyses did not reveal any significant difference between the wild-type and *Lrig*-null MEF lines with regard to their aerobic or anaerobic responses, as measured by the oxygen consumption rate (OCR) and extracellular acidification rate (ECAR), respectively (Supplementary Fig. 1f, g).

**Lrig proteins promote adipogenesis in vitro.** To investigate the role of Lrig proteins in adipogenesis in vitro, wild-type and *Lrig*-null MEF lines were treated with an adipogenic cocktail consisting of the glucocorticoid dexamethasone, the cAMP diesterase inhibitor 3-isobutyl-1-methylxanthine, insulin, and the PPARγ activator rosiglitazone. After nine days of treatment, adipocytic transformation was assessed by Oil Red O staining. Three out of four wild-type MEF lines were clearly able to transform into adipocytes in response to the adipogenic cocktail, whereas all the *Lrig*-null MEF lines studied showed impaired adipogenesis. However, because the adipogenic potential was highly variable

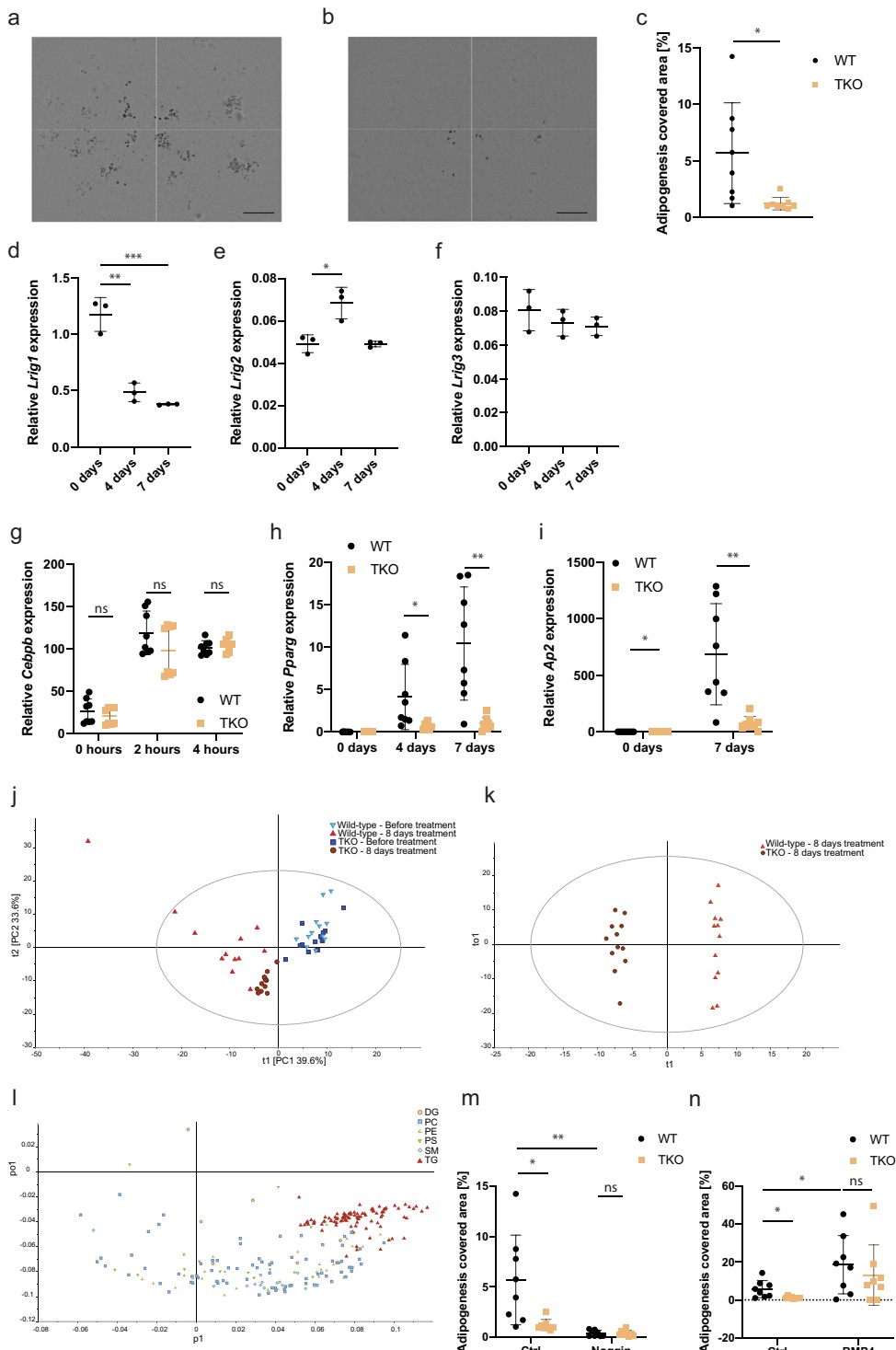

among the wild-type MEF lines, we included four additional biological replicates of both wild-type and *Lrig*-null lines in the analysis. Taken together, the analysis of the eight wild-type and eight *Lrig*-null MEF lines clearly showed that compared to wild-type MEFs, *Lrig*-null MEFs had deficient adipocyte differentiation (Fig. 1a–c). During the adipogenic process of wild-type MEFs, *Lrig1* was downregulated and *Lrig2* transiently upregulated, whereas *Lrig3* did not show any significant changes (Fig. 1d–f, Supplementary Fig. 2a–f). It has been reported that the induction of adipogenesis results in the rapid induction of *Cebpb* expression, followed by the induction of *Pparg* expression, and finally, in differentiated adipocytes, *Ap2* expression[37–40]. Gene expression

analyses via quantitative reverse transcription-PCR (qRT-PCR) revealed that the induction of *Cebpb* was not diminished in *Lrig*-null MEFs compared to in wild-type MEFs (Fig. 1g). However, the induction of *Pparg* and *Ap2* was severely impaired in *Lrig*-null MEFs compared to in wild-type MEFs (Fig. 1h, i). Although Oil Red O staining is a well-established method to visualize triglyceride-containing adipocytes, we wanted to investigate the biochemical changes in lipid composition that were associated with adipogenesis in our experimental system. To this end, we performed lipid profiling of the MEFs by liquid chromatography coupled with tandem mass spectrometry prior to the adipogenic treatment as well as eight days after the induction of adipogenesis.

**Fig. 1 Lrig proteins regulate adipogenesis of MEFs in vitro.** Wild-type (WT) and *Lrig*-null (TKO) MEFs were treated with an adipogenic cocktail as described in the Methods section for the indicated times. **a–c** Adipogenesis of wild-type and *Lrig*-null MEF lines. Wild-type and *Lrig*-null MEFs were treated with the adipogenic cocktail for nine days followed by the quantification of adipocytes via Oil Red O staining. Shown are representative images of wild-type cells with 6% covered area (**a**) and *Lrig*-null cells with 1.7% covered area (**b**) (scale bar, 0.6 mm) and the quantifications of percentage area coverage for the Oil Red O stained biological replicates ($n = 8$ per genotype) (**c**). **d–i** Relative mRNA expression levels of *Lrig1* (**d**), *Lrig2* (**e**), *Lrig3* (**f**), *Cebpb* (**g**), *Pparg* (**h**), and *Ap2* (**i**). Cells were treated as in a-c. At the indicated time points after induction, the cells were lysed and analyzed by quantitative RT-PCR. Expression was normalized to the reference gene *Rn18s*. Shown in d-f are wild-type cells only, whereas both wild-type and *Lrig*-null cells are shown in g–i, as indicated, for eight biological replicates per genotype. **j–l** Lipidomic analyses. Lipids were extracted from wild-type and *Lrig*-null MEFs before or after eight days of treatment with the adipogenic cocktail. Lipid analysis was then performed by liquid chromatography coupled with tandem mass spectrometry. Each symbol represents one experimental replicate; shown are the results of three biological replicates per genotype with four experimental replicates each. **j** PCA score plots of all samples, labeled according to sample category. The variation explained by PC1 and PC2 was 39.6% and 33.6%, respectively. **k** The score plot of the OPLS-DA model built from the lipid profiles of the 8-day samples to determine the maximal variance between the wild-type and *Lrig*-null sample groups. **l** The corresponding loading plot explaining the contributions of different lipid species to the OPLS-DA model, indicating that triglycerides (TGs) (red triangles) in the wild-type samples were highly enriched compared to in the *Lrig*-null samples. The lipids are labeled according to lipid class (DG: diacylglycerol, PC: phosphatidylcholine, PE: phosphatidylethanolamine, PS: phosphatidylserine, SM: sphingomyelin, TG: triacylglycerol). **m, n** Role of BMP for adipogenesis in vitro. Wild-type and *Lrig*-null MEFs were treated as in a, without (Ctrl) or with the addition of 100 ng/ml of the BMP inhibitor noggin (Noggin) (**m**), or without (Ctrl) or with the addition of 50 ng/ml of BMP4 (BMP4) (**n**). Adipogenesis was scored through Oil Red O staining as described under **a**. In **c-i**, **m** and **n** the means of the biological replicates (**c**, **g-i**, **m** and **n**, $n = 8$ per genotype; **d-f**, $n = 3$) are shown by horizontal lines, and the means of the individual biological replicates analyzed by three experimental repeats are shown by dots and squares. Error bars represent the standard deviations of the means of the biological replicates. $^{ns}P > 0.05$, $*P < 0.05$, $**P < 0.01$ (Student's *t*-test).

In total, 244 putative lipids were quantified. A principal component analysis (PCA) of all samples did not reveal any apparent difference between the wild-type and *Lrig*-null MEFs prior to adipogenesis induction (Fig. 1j). Eight days after induction of adipogenesis, both the wild-type and the *Lrig*-null MEF lines separated distinctly from the untreated control MEFs. At this time point, separation was also evident between wild-type and *Lrig*-null MEFs. Thus, all the MEF lines responded to the adipogenic stimulus by altering their lipid composition; however, the wild-type and *Lrig*-null MEFs did so in different ways. In fact, a supervised orthogonal projections to latent structures discriminant analysis (OPLS-DA) plot completely separated the treated wild-type samples from the treated *Lrig*-null samples (Fig. 1k). The corresponding loadings plot, using the lipid classes, showed that the main contributors to the separation between the treated wild-type and treated *Lrig*-null samples were the triacylglycerides, of which the majority showed higher levels in wild-type MEFs than in *Lrig*-null MEFs (Fig. 1l).

TGF-β and BMP signaling pathways have been reported to play important roles in adipogenesis in vitro and in vivo[41]. Accordingly, the BMP inhibitor noggin was able to inhibit the adipogenesis cocktail-induced adipogenesis of wild-type MEFs (Fig. 1m). Conversely, a high dose of BMP4 (50 ng/ml) was able to greatly enhance the cocktail-induced adipogenesis rate of wild-type MEFs and, intriguingly, rescued the adipogenesis deficiency of the *Lrig*-null MEFs (Fig. 1n).

**Lrig-null MEFs show impaired BMP signaling.** To investigate the role of the LRIG proteins in BMP signaling, we used a BMP-responsive element-driven luciferase reporter gene assay and analyzed the phosphorylation levels of Smad1/5 by fluorescent immunocytochemistry and Western blotting. First, we transiently transfected the wild-type and *Lrig*-null MEFs with the BMP reporter *pGL3-BRE-luciferase* and then stimulated them with different concentrations of BMP4. In this assay, the *Lrig*-null MEF lines showed a lower sensitivity to BMP4 than the wild-type MEF lines (Fig. 2a). Similarly, the pSmad1/5 analysis showed that the *Lrig*-null MEFs had a lower BMP4 sensitivity than the wild-type MEFs (Fig. 2b); however, the maximal pSmad1/5 response did not appear to differ between the wild-type and *Lrig*-null MEFs in this assay. Additionally, the *Lrig*-null MEFs showed a reduced sensitivity for BMP6 (Fig. 2c), whereas the sensitivity for BMP9/GDF2 was similar between the wild-type and the *Lrig*-null

MEF lines (Fig. 2d). Compared with wild-type MEFs, *Lrig1* and *Lrig3* single-knockout MEF lines (Supplementary Fig. 2g, h) showed an apparently unaltered BMP4 sensitivity (Fig. 2e, f). BMPs are also able to activate noncanonical BMP signaling, which includes the activation of MAPK signaling cascades[21,22]. Intriguingly, the *Lrig*-null MEFs showed a reduced sensitivity for BMP4 when noncanonical phosphorylation of p38 was analyzed (Fig. 2g–i); however, the detection of increased p38 phosphorylation levels required higher BMP4 concentrations than the detection of increased pSmad1/5.

To investigate whether the BMP signaling deficiency of the *Lrig*-null MEF lines was the result of reduced expression of one or several of the BMP receptors, the BMP receptor levels were analyzed at the transcript level by RNA sequencing (RNAseq) and at the protein level by Western blotting. The RNAseq analysis revealed no significant difference in the levels of the different BMP receptor transcripts between the wild-type and *Lrig*-null MEF lines (Fig. 2j). Accordingly, Acvr1 and Bmpr2 showed similar protein expression levels in wild-type and *Lrig*-null MEFs when analyzed through Western blotting (Supplementary Fig. 4a–c). In addition, there were no significant differences in the transcript levels of Bmp ligands, signaling mediators, responsive genes, or Tgf-β receptors (Fig. 2j).

**LRIG1 and LRIG3 rescue BMP signaling in Lrig-null MEFs.** To investigate whether individual *LRIG* alleles could rescue the *Lrig*-null phenotype, an *Lrig*-null MEF line was transduced with the inducible human alleles *LRIG1*, *LRIG2*, or *LRIG3*, with an empty vector serving as a control (Supplementary Fig. 2i–k). As assessed by flow cytometry, a majority of the transduced cells expressed LRIG1 or LRIG3 after induction, whereas the lower expression level of LRIG2 made it difficult to determine the fraction of LRIG2-positive cells with this method (Supplementary Fig. 2l–o). Intriguingly, the induction of *LRIG1* or *LRIG3* expression rescued the canonical BMP sensitivity phenotype of the *Lrig*-null MEFs, whereas the induction of *LRIG2* expression, or vector control, did not (Fig. 3a–d). Interestingly, noncanonical BMP signaling through p38 and Jnk phosphorylation was only rescued by LRIG1 and not by LRIG3 (Fig. 3e, f, g–j). Increased phosphorylation of Erk was not observed under the BMP stimulation-protocol used (Fig. 3e, f, k, l). To investigate whether LRIG1 and LRIG3 could also rescue the adipogenesis deficiency of *Lrig*-null MEFs, *LRIG1*- and *LRIG3*-inducible MEFs were analyzed. Clearly, the induced

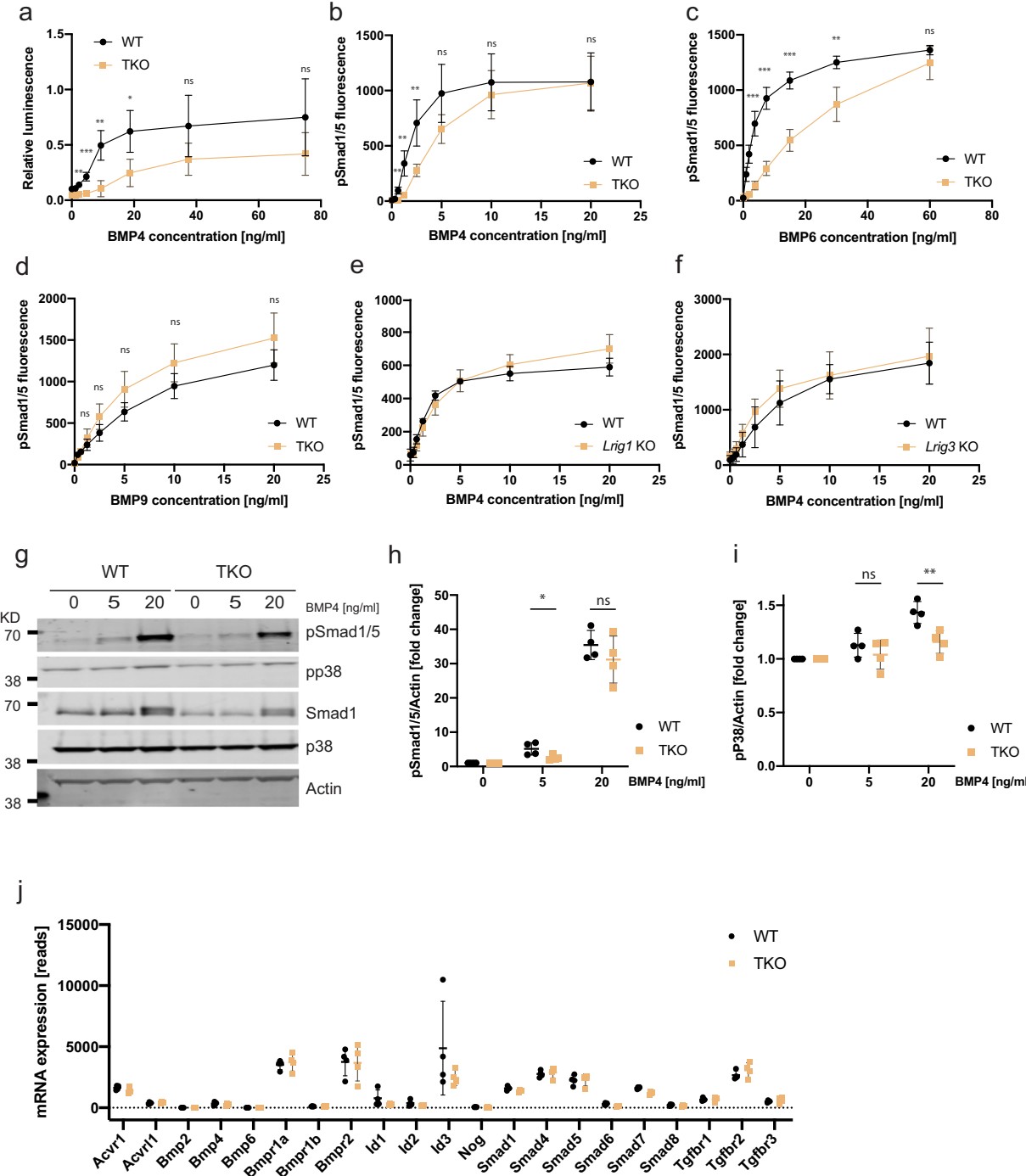

**Fig. 2 Lrig-null MEFs show impaired BMP signaling without any apparent changes in the expression of receptors or signaling mediators. a** Wild-type (WT) and *Lrig*-null (TKO) MEFs expressing the BMP reporter plasmid *pGL3-BRE-luciferase* were treated with the indicated concentrations of BMP4 for three hours. Thereafter, the cells were lysed, and the luciferase activity was analyzed and normalized to the control. **b–d** Wild-type and *Lrig*-null MEFs were stimulated with various concentrations of BMP4 (**b**), BMP6 (**c**), or BMP9 (**d**) for one hour followed by immunocytofluorescence analysis of nuclear phospho-Smad1/5 (pSmad1/5). **e, f** Wild-type and *Lrig1*-null MEFs (**e**) or wild-type and *Lrig3*-null MEFs (**f**) were stimulated with various concentrations of BMP4 for one hour followed by nuclear pSmad1/5 analysis. **g–i** Western blot analyses of canonical BMP4 signaling through pSmad1/5 and noncanonical BMP signaling through phosphorylated p38 (pp38). Wild-type and *Lrig*-null MEFs were stimulated with the indicated concentrations of BMP4 for one hour followed by cell lysis and Western blot analysis. Uncropped blots are shown in Supplementary Fig. 3. **g** Representative Western blots showing pSmad1/5, pp38, total Smad1, total p38, and the loading control actin. **h** Quantification of the pSmad1/5/actin ratios. **i** Quantification of the pp38/actin ratios. **j** Gene expression levels were analyzed in wild-type (WT) and *Lrig*-null (TKO) MEFs via RNA sequencing (RNAseq). The apparent number of RNAseq reads for respective gene is indicated. All the values in **a–f**, **h** and **i** represent the means of four biological replicates that were analyzed by three experimental repeats each. **j** The values represent the means of four biological replicates that were analyzed once. Error bars represent the standard deviations of means from four biological replicates. $^{ns}P > 0.05$, $^{*}P < 0.05$, $^{**}P < 0.01$, $^{***}P < 0.001$ (Student's *t*-test).

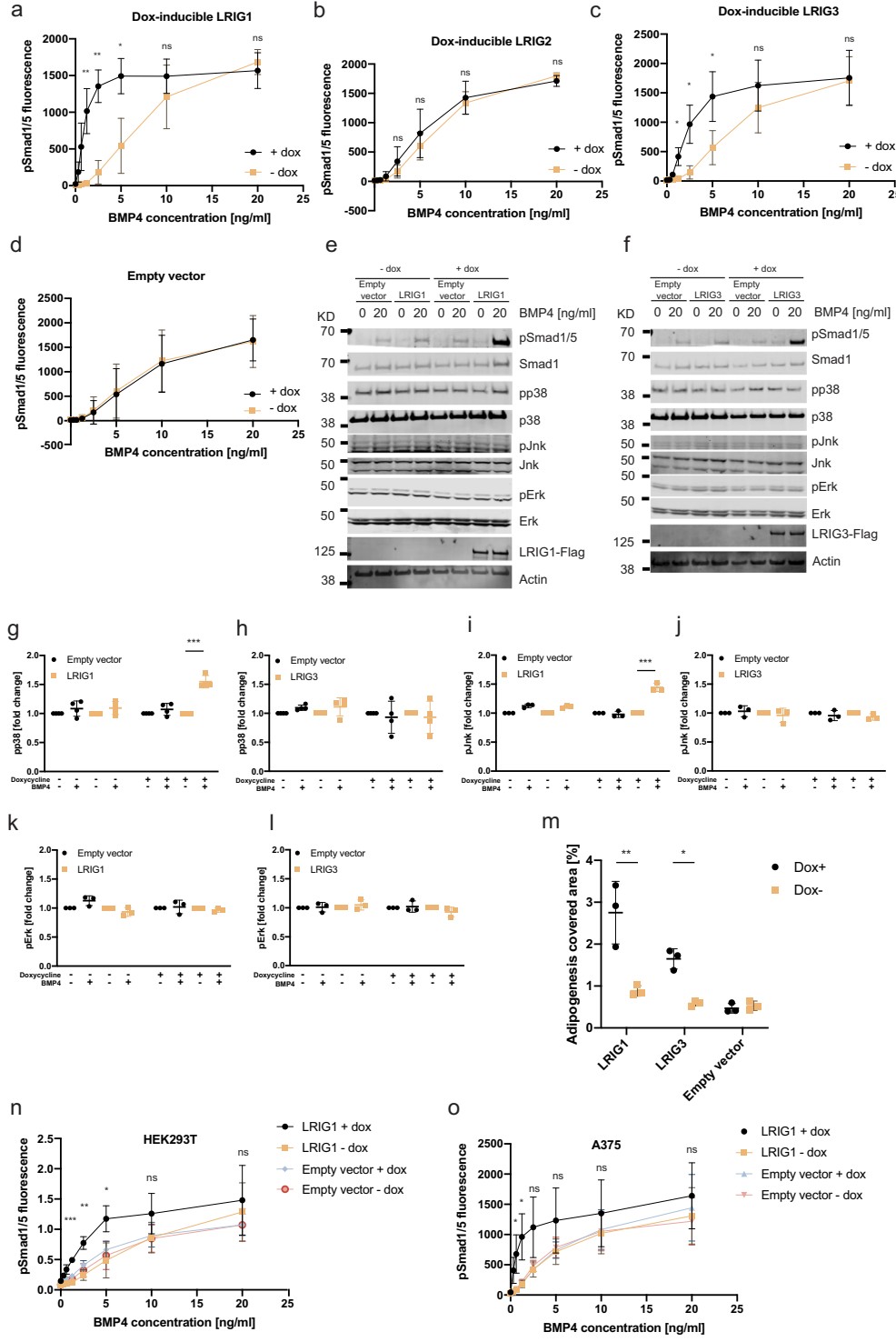

expression of LRIG1 or LRIG3 enhanced the adipogenesis rate of the MEF lines (Fig. 3m). To investigate whether the observed sensitizing effects of LRIG proteins on BMP signaling were restricted to MEFs, or if it was a more general phenomenon, the *LRIG1*-inducible human HEK293T and A375 cell lines were analyzed. Clearly, when *LRIG1* expression was induced in HEK293T or A375 cells, both cell lines showed a greatly enhanced sensitivity to low concentrations of BMP4 (Fig. 3n, o). To compare the BMP4-sensitizing potencies of the human LRIG proteins, different levels of LRIG1, LRIG2, or LRIG3 were induced by titrating the specific transcription-inducer doxycycline. Then, the correlations between the LRIG protein expression

levels and the BMP4-induced pSmad1/5 levels were determined. This correlation analysis revealed that LRIG1 and LRIG3 were approximately equally potent in sensitizing the *Lrig*-null MEFs to BMP4; as expected, LRIG2 showed a negligible effect on BMP4 signaling (Fig. 4a). Next, we performed a structure-function analysis of the relationships between different protein domains and the BMP-sensitizing function of LRIG1. To this end, *Lrig*-null MEFs were transiently transfected with different amounts of expression vectors encoding a green fluorescent protein (GFP) -tagged full-length LRIG1, a GFP-tagged LRIG1 variant that lacked the cytosolic tail (LRIG1-Δcyto), or GFP only (as transfection control). Because of the poor transfection

**Fig. 3 Ectopic LRIG1 or LRIG3 resensitize *Lrig*-null cells to BMP4. a–m** *Lrig*-null MEFs were transduced with doxycycline-inducible *LRIG1*, *LRIG2*, or *LRIG3* constructs or with empty vector as a noninducible control. LRIG protein expression was not induced or induced by treatment of the cells with 100 ng/ml (**a–d**, **m–o**) or 500 ng/ml (**e–l**) doxycycline for 24 h followed by stimulation of the cells with different concentrations of BMP4 for one hour (**a–l**, **n**, **o**) or with adipogenic cocktail for ten days (**m**). **a–d** Immunofluorescence analyses of nuclear pSmad1/5 in cells not induced (−dox) or induced (+dox) to express *LRIG1* (**a**), *LRIG2* (**b**), or *LRIG3* (**c**). The empty vector served as a negative control for doxycycline treatment (**d**). **e–l** Western blot analyses of canonical (pSmad1/5) and noncanonical (pp38, pJnk, and pErk) BMP4 signaling. *LRIG1*- or *LRIG3*-inducible MEFs were induced, or not induced, with doxycycline followed by stimulation with 0 or 20 ng/ml of BMP4 for one hour. Thereafter, the cells were lysed, and the lysates were analyzed by Western blotting. **e, f** Representative Western blots showing pSmad1/5, total Smad1, pp38, total p38, pJnk, total Jnk, pErk, total Erk, LRIG1-FLAG (**e**), LRIG3-FLAG (**f**), and the loading control actin. Uncropped blots are shown in Supplementary Fig. 5. **g–l** Quantification of pp38 (**g**, **h**), pJnk (**i**, **j**), and pErk (**k**, **l**) normalized to actin in *LRIG1*-inducible (**g**, **i**, **k**) or *LRIG3*-inducible (**h**, **j**, **l**) MEFs. Plotted values in **a–d** represent means from three biological replicates, each with three experimental repeats. Error bars represent the standard deviations of means from three biological replicates. **g, h** Shown are four experimental repeats using an *LRIG1*- or *LRIG3*-inducible MEF line. Error bars show the standard deviations of the four means. **i–l** Shown are three experimental repeats using an *LRIG1*- or *LRIG3*-inducible MEF line. Error bars show standard deviations of the four means. **m** *LRIG1* or *LRIG3* expression was induced or not in *Lrig*-null MEFs with doxycycline followed by treatment of the cells with the adipogenic cocktail for ten days and quantification of adipocytes via Oil Red O staining. Shown are quantifications of the percentage area coverage for the Oil Red O stained biological replicates (*n* = 3 per genotype and treatment). **n, o** *LRIG1* expression was induced or not in *LRIG1*-inducible HEK293T cells (**n**) and A375 cells (**o**) via the treatment of cells with doxycycline for 24 h, followed by stimulation with different concentrations of BMP4 for one hour. Immunofluorescence analyses of nuclear pSmad1/5 in HEK293T cells (**n**) or A375 cells (**o**) that were not induced (-dox) or induced (+dox) to express *LRIG1*, with empty vector serving as a noninducible control. Plotted values in **n** and **o** represent the means from three independent experiments, performed as triplicates using one biological replicate of each cell line. Error bars represent standard deviations from three means. $^{ns}P > 0.05$, $^*P < 0.05$, $^{**}P < 0.01$, $^{***}P < 0.001$ (Student's *t*-test).

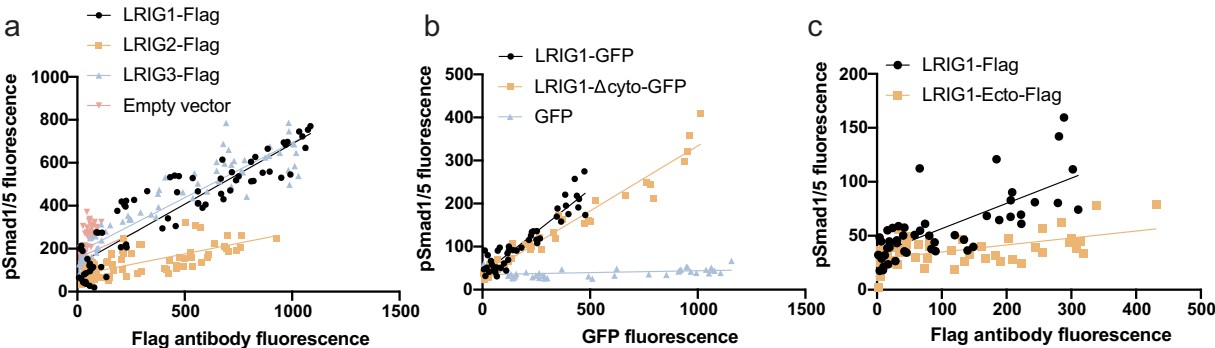

**Fig. 4 LRIG1 and LRIG3 promote BMP signaling. a** Different LRIG protein expression levels were induced in *LRIG*-inducible MEFs by treating the cells with different concentrations of the inducer doxycycline. Thereafter, the cells were stimulated with 2.5 ng/ml BMP4 for one hour followed by coimmunocytofluorescence analyses of nuclear pSmad1/5 and the FLAG-epitope present as a tag on the induced LRIG proteins. Correlation plots of phosphorylated Smad1/5 versus the FLAG-LRIG protein expression levels are shown. Fitted lines indicate the linear relationship with pSmad1/5 for each LRIG protein. Pearson's correlation coefficients for the respective genes were LRIG1, 0.9162; LRIG2, 0.6939; LRIG3, 0.9251; and empty vector, 0.1659. Shown are three experimental repeats using three biological replicates. **b** *Lrig*-null MEFs were transfected with a full-length LRIG1-GFP fusion protein (LRIG1-GFP), an LRIG1-GFP fusion protein variant lacking the cytosolic domain of LRIG1 (LRIG1-Δcyto-GFP), or empty vector (GFP) as a transfection control. Thereafter, the cells were stimulated with 20 ng/ml of BMP4 for 20 min followed by coimmunocytofluorescence analyses of nuclear pSmad1/5 and the green fluorescence from GFP fusion proteins or control GFP. The correlation plots between pSmad1/5 and GFP fluorescence are shown. The fitted lines indicate the linear relationship to pSmad1/5 for the respective construct. Pearson's correlation coefficients for the respective constructs were as follows: full length LRIG1, 0.8943; LRIG1-Δcyto, 0.9663; GFP control, 0.2564. Shown are two experimental repeats using three biological replicates. **c** *Lrig*-null MEFs were transfected with different amounts of expression vectors encoding FLAG-tagged full-length LRIG1 (LRIG1) or FLAG-tagged LRIG1 ectodomains (LRIG1-ecto). Thereafter, the cells were stimulated with 20 ng/ml of BMP4 for 20 min followed by coimmunocytofluorescence analyses of nuclear pSmad1/5 and the FLAG-epitope. Shown are the correlation plots between pSmad1/5 and FLAG-LRIG expression levels. Fitted lines indicate the linear relationship between pSmad1/5 and the respective FLAG-LRIG construct. Pearson's correlation coefficients for the respective constructs were as follows: full-length LRIG1, 0.7393; and LRIG1-ecto, 0.5287. Shown are two experimental repeats using three biological replicates.

efficiencies of our MEF lines, the background signals from the majority of nontransformed *Lrig*-null cells imposed an analytical problem. To resolve this problem, we changed the stimulation protocol from 2.5 ng/ml BMP4 for 60 min to 20 ng/ml BMP4 for 20 min. This modified protocol enabled us to monitor the pSmad1/5 signals among the minority of *LRIG1*-transformed MEFs while keeping the background signals from the majority of nontransformed *Lrig*-null MEFs to a minimum. By correlating the BMP4-induced pSmad1/5 responses with the expression levels of the transfected LRIG1 or LRIG1-Δcyto proteins, it was revealed that full-length LRIG1 and LRIG1-Δcyto were approximately equally potent in promoting BMP4 signaling in the *Lrig*-null MEFs (Fig. 4b). The BMP4-sensitizing function of full-length

LRIG1 was also compared with the isolated ectodomain of LRIG1, that is, LRIG1 lacking its transmembrane and cytosolic domains. Apparently, the LRIG1 variant lacking the transmembrane and cytosolic domains lost its BMP4-sensitizing function (Fig. 4c). Thus, the cytosolic tail, but not the transmembrane domain, was dispensable for the BMP-sensitizing function of LRIG1 in the context studied.

**TGF-β and RTK-MAPK signaling pathways appear to be Lrig-independent in MEFs.** Because the TGF-β and BMP signaling pathways share many common features and because RTK signaling has been reported to be regulated by LRIG proteins, we

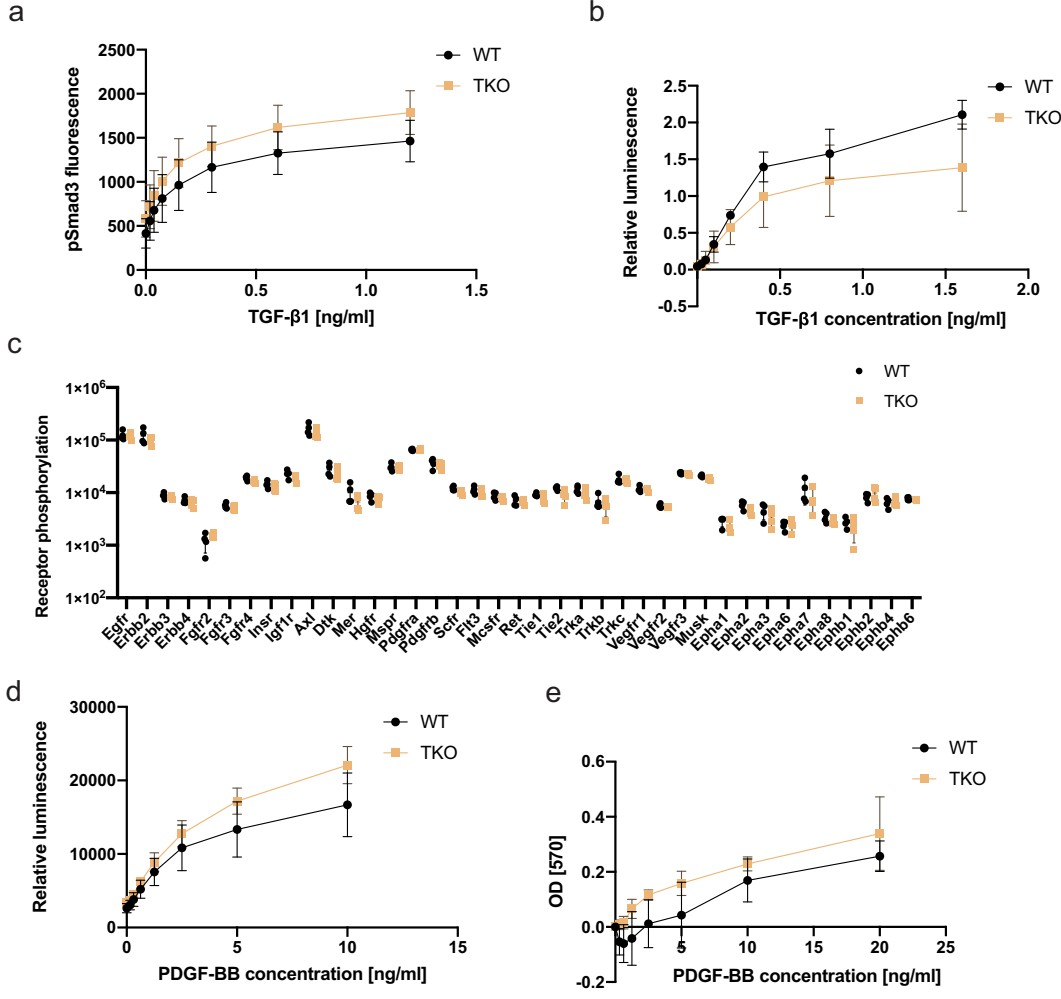

**Fig. 5 TGF-β and RTK signaling appear unaltered in *Lrig*-null MEFs. a** Wild-type (WT) and *Lrig*-null (TKO) MEFs were stimulated with various concentrations of TGF-β1 for one hour and then analyzed for nuclear phospho-Smad3 (pSmad3) by an immunocytofluorescence assay. **b** Wild-type and *Lrig*-null MEFs were transiently transfected with the TGF-β reporter plasmid *p(CAGA)₁₂MLP-Luc* followed by treatment of the cells with the indicated concentrations of TGF-β1 for three hours. Cell lysates were then analyzed for luciferase activity. Shown is the relative luminescence on an arbitrary scale. **c** Wild-type and *Lrig*-null MEFs, cultivated under standard cell culture conditions in 10% FBS, were lysed and analyzed for RTK phosphorylation levels with a phospho-RTK array kit (R&D Systems). **d** Wild-type and *Lrig*-null MEFs stably expressing the MAPK reporter gene *ELK1/SRF-luc* were treated with different concentrations of PDGF-BB for four hours and then lysed and analyzed for luciferase activity. **e** Wild-type and *Lrig*-null MEFs were cultivated in FBS-free medium containing different amounts of PDGF-BB for 48 h. Thereafter, the relative cell numbers were determined using an MTT assay. Shown are the OD values of treated MEFs normalized to those of untreated MEFs. **a**, **b**, **d** and **e** The plotted values represent the means of four biological replicates, each with three experimental repeats. **c** The plotted values represent the means of four biological replicates, each with one experimental repeat. Error bars represent the standard deviations of the means from four biological replicates.

also compared TGF-β and RTK signaling between wild-type and *Lrig*-null MEFs. TGF-β signaling was assessed using a TGF-β-responsive element-driven luciferase reporter, *p(CAGA)₁₂MLP-Luc*, assay[42] or by analyzing the phosphorylation levels of Smad3 by fluorescent immunocytochemistry. Neither of these analyses revealed any significant difference between wild-type and *Lrig*-null MEFs with regard to their TGF-β1 responses (Fig. 5a, b). To investigate the role of Lrig proteins in steady-state RTK signaling levels in MEFs under standard cell culture conditions with 10% FBS, a phospho-RTK array was used. Surprisingly, there was no apparent difference in the specific RTK phosphorylation levels between the wild-type and *Lrig*-null MEFs under the standard cell culture conditions employed (Fig. 5c). To further analyze the role of LRIG proteins in RTK signaling, a wild-type MEF line carrying floxed *Lrig* alleles was stably transduced with the MAPK reporter gene *ELK1/SRF-luc*. Thereafter, the stably transduced MAPK reporter MEFs were transduced with Cre recombinase or a

control vector to generate four independent MAPK reporter *Lrig*-null MEF lines together with four MAPK reporter wild-type MEF lines (Supplementary Fig. 4d–f). When these MEF lines were treated with different concentrations of platelet-derived growth factor (PDGF) -BB, luciferase expression was induced; however, there was no apparent difference in the sensitivity to PDGF-BB between the wild-type and *Lrig*-null MEFs (Fig. 5d). Additionally, the dose-response of PDGF-BB-induced proliferation was similar between the wild-type and *Lrig*-null MEF lines (Fig. 5e).

**The *LRIG* homolog *sma-10* regulates lipid metabolism in *C. elegans*.** In *C. elegans*, LRIG/SMA-10 is reported to promote normal body size through BMP signaling[12]. Given that several BMP mutants have been reported to be defective in lipid homeostasis[34,35], we hypothesized that defective lipid homeostasis could be a general trait of BMP mutants, including

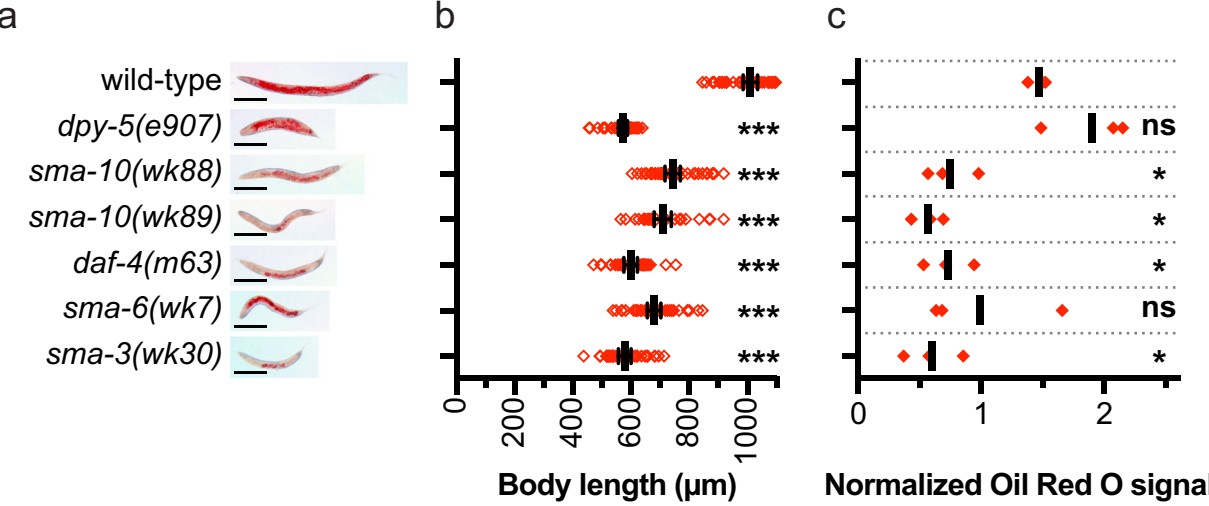

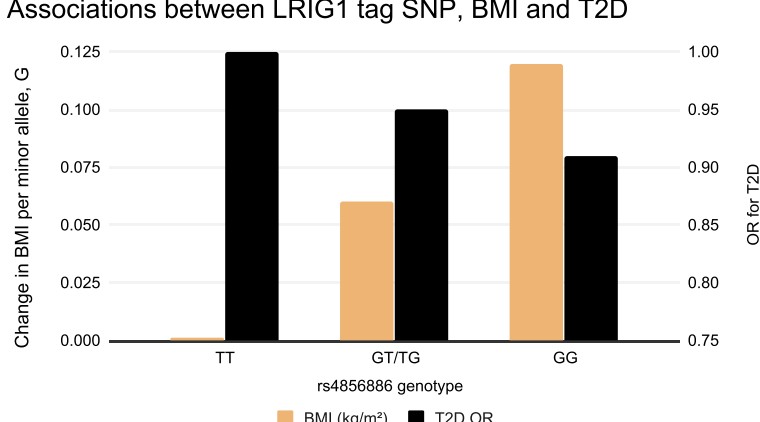

**Fig. 6 *LRIG/sma-10* and other BMP pathway genes promote lipid accumulation in lipid-storing cells in *C. elegans*, and *LRIG1* SNPs predict BMI and type 2 diabetes in humans. a** Representative whole-body images of Oil Red O-stained adult hermaphrodite worms. Scale bars, 200 μm. **b** Adult body lengths of wild-type animals (*n* = 46), *dpy-5(e907)* (*n* = 37), *sma-10(wk88)* (*n* = 41), *sma-10(wk89)* (*n* = 32), *daf-4(m63)* (*n* = 32), *sma-6(wk7)* (*n* = 41), and *sma-3 (wk30)* (*n* = 37). The body length of each individual animal is plotted as a red square. Solid lines and error bars indicate the means and 95% confidence intervals, respectively. The order of the dot plots, from top-to-bottom, is the same as that for the images in **a**. **c** Oil Red O signal intensities from three independent experiments. Each experiment was normalized to its combined mean signal intensity across all genotypes. For each genotype, solid, red squares indicate the mean normalized signal in each independent experiment. Solid lines indicate the combined means from three experiments. The order of the dot plots, from top-to-bottom, is the same as those for the images in a. Statistical significance versus wild-type was determined with multiplicity-adjusted *P*-values, calculated using Holm-Sidak multiple comparisons tests. **P* < 0.05, ****P* < 0.001. **d** Plot illustrating the difference in predicted BMI (least square means, LSMs) across the genotypes of rs4856886 (minor allele = G, major allele = T) and odds ratio for type 2 diabetes (T2D) across genotypes. The *x*-axis represents the rs4856886 genotypes compared. The *y*-axis on the left represents the difference in BMI LSMs per single minor allele across the genotypes, while the y-axis on the right represents the odds ratios for T2D. In this study, the minor allele was associated with an increase in BMI and a lower odds of T2D risk.

mutants for *LRIG/sma-10*. As a BMP pathway-independent control for short body length, we included the cuticle collagen mutant *dpy-5(e907)* in our analysis[43]. Both body length and fat accumulation were assessed in wild-type and mutant worms (Fig. 6a–c). As expected, compared to wild-type worms, all of the *dpy-5*, *sma-10*, *daf-4*, *sma-6*, and *sma-3* mutant worms showed a reduced body length (Fig. 6a, b). Intriguingly, compared to lipid accumulation in wild-type worms, lipid accumulation was reduced in both the *LRIG/sma-10* mutants *wk88* and *wk89*, in the *ACVR2A* and *ACVR2B* homolog mutant *daf-4(m63)* and in the *SMAD1* homolog mutant *sma-3(wk30)* (Fig. 6a, c). Compared to the wild-type, also the *BMPR1A/sma-6(wk7)* mutants showed an apparent, although nonsignificant (*p* = 0.14), reduction in lipid

accumulation. For unknown reasons, the *sma-6(wk7)* series showed a higher variance than the other BMP mutants, which might have contributed to the lack of significance for this series. Despite being short, the *dpy-5(e907)* mutants had normal levels of lipid deposits in lipid-storing intestinal cells. Short body length per se thereby appears uncoupled from fat accumulation in somatic tissue of *C. elegans*. Hence, we conclude that *LRIG/sma-10* promotes lipid accumulation in postmitotic tissue of *C. elegans*, likely through BMP signaling and independent of body size regulation.

**Human *LRIG1* variants are associated with an altered risk of type 2 diabetes and with BMI and adipocyte morphology.** To

investigate possible associations between *LRIG* gene variants and human metabolism and metabolic disease, data from the UK Biobank were analyzed ($n = 398,810$, Supplementary Table 3 for participant characteristics). Here, we identified nine variants at *LRIG1* that were strongly associated with BMI ($P < 5 \times 10^{-8}$) (Supplementary Table 4); these signals have also been reported by others as a part of a larger meta-analysis conducted while this study was in progress[44]. In our analyses, each copy of the minor allele of *rs4856886* (G) (tag SNP with the strongest effect) increased the BMI by ~0.05 kg/m$^2$ (Fig. 6d); this phenomenon was largely attributed to genetic variation (Supplementary Table 5). These variants were also associated with a decreased risk of diabetes and adjusting for BMI strengthened these associations (Fig. 6d; Supplementary Table 6). Secondary signals for many other metabolic traits, including plasma triglyceride levels (Supplementary Table 7), were also observed, suggesting a metabolically favorable phenotype in people carrying the BMI-associated *LRIG1* alleles. The results of association with liver fat percentage were not significant (Supplementary Table 8), which might be attributable to the low statistical power given the relatively small sample size ($n = 3,192$, compared to $n = 398,810$ for other analyses). We hypothesized that the observed relationships may be attributed to a metabolically favorable adipose tissue phenotype, which is typically observed in adipose tissue comprised of many small adipocytes, i.e., hyperplastic adipose morphology[45]. Thus, for the two strongest *LRIG1* signals from UK Biobank (*rs4856886* and *rs9840088*), we tested associations with adipocyte size in adults from the GENiAL cohort ($n = 948$, Supplementary Table 9). In these analyses, the BMI increasing (type 2 diabetes risk decreasing) alleles at *rs4856886* ($P = 0.039$) and *rs9840088* ($P = 0.014$) were associated with adipose hyperplasia. For *rs9840088*, the mean adipose morphology values were +7.5 picolitres for the common A allele and −9.2 picolitres for the minor C allele ($P = 0.026$ by analysis of covariance, adjusted for age).

## Discussion

Lipid metabolism is central to energy homeostasis at both the cellular and organismal levels. Here, we found that the LRIG proteins function as regulators of lipid metabolism by regulating BMP signaling in several different biological systems, including adipocyte differentiation of mouse fibroblasts and in lipid accumulation in *C. elegans*; in addition, we found that human *LRIG1* gene variants were associated with a decreased risk of type 2 diabetes, an increased BMI, and altered adipocyte morphology. Collectively, these observations show that LRIG proteins function as evolutionarily conserved regulators of BMP signaling and lipid metabolism and have important implications for human metabolic health.

LRIG proteins regulated lipid accumulation and adipogenesis of MEFs in response to adipogenic stimuli. The former was shown by a reduced triglyceride accumulation, whereas the latter was suggested by a reduced number of equally sized Oil Red O-positive cells and impaired induction of the adipocyte markers *Pparg* and *Ap2* among the *Lrig*-null MEFs. Furthermore, the associations between *LRIG1* gene variants and adipocyte numbers in humans suggests that LRIG proteins might also regulate adipogenesis in vivo in humans. However, although the lipid profiles of undifferentiated MEFs were indistinguishable between the wild-type and *Lrig*-null cells, our experiments did not address whether LRIG proteins could regulate metabolism in differentiated adipocytes. In this regard, it was intriguing that the *LRIG/sma-10* mutant nematodes also showed a lipid accumulation defect, although *C. elegans* lack dedicated adipocytes. Thus, further investigations will reveal whether mammalian LRIG proteins, in addition to regulating adipogenesis, also regulate lipid

metabolism in differentiated energy-regulating cells such as adipocytes, hepatocytes, or skeletal muscle cells.

The sole *C. elegans* LRIG homolog, SMA-10, regulates body size by regulating BMP signaling. Here, we showed that the BMP-promoting function of SMA-10 is conserved in human LRIG1 and LRIG3, which promoted BMP signaling by increasing signaling strength at low BMP4 and BMP6 ligand concentrations. These results show that the BMP signal-regulating function of the hypothesized common protein ancestor of nematode SMA-10 and mammalian LRIG proteins seems to be retained in human LRIG1 and LRIG3, but not in LRIG2. This finding is consistent with a recent whole genome CRISPR-Cas9 phenotypic screen for regulators of BMP signaling in HEK293 cells[46]. In this dataset, LRIG1 and LRIG3 were among the 170 significant activators of BMP2-induced signaling, whereas LRIG2 was not (https://orcs.thebiogrid.org/Screen/172). Our demonstration that the human LRIG1 cytosolic domain was dispensable for its BMP signal-promoting function is consistent with the fact that in contrast to the mammalian LRIG proteins, *C. elegans* SMA-10 lacks a prominent cytosolic domain but still promotes BMP signaling. Intriguingly, the mammalian LRIG proteins showed a striking specificity with regard to the BMP pathways that they regulated. Thus, BMP4 and BMP6 signaling was strongly dependent on LRIG proteins, whereas BMP9 signaling was not. This discrimination between the BMP4/6 and BMP9 pathways may indicate possible molecular targets for the mammalian LRIG proteins. BMP4, BMP6, and BMP9 share the same type 2 receptors; however, BMP4 and BMP6 specifically interact with the type 1 receptors BMPR1A (ALK-3) and BMPR1B (ALK-6), whereas BMP9 specifically interacts with ACTRL1 (ALK-1). Accordingly, it can be speculated that LRIG1 and LRIG3 may regulate the BMP type 1 receptors BMPR1A and/or BMPR1B but not the type 2 receptors or the type 1 receptor ACTRL1. Mammalian LRIG proteins specifically regulating type 1 BMP receptors would be in line with the demonstration that *C. elegans* SMA-10 is required for the proper trafficking of SMA-6, a type 1 BMP receptor, but not for the trafficking of DAF-4, the type 2 receptor regulating body size[13]. Curiously, we found that LRIG1, but not LRIG3, could rescue noncanonical BMP signaling through p38 and Jnk MAPKs. The molecular basis for these specific functions of LRIG proteins remains to be elucidated.

We further examined whether genetic variation in *LRIG1* was associated with risk of type 2 diabetes, BMI, and adipocyte morphology among humans. Intriguingly, we identified diabetes-preventing *LRIG1* alleles that were strongly associated with an increased BMI and hyperplastic adipose morphology, e.g. many small adipocytes given total body fat. All the identified SNPs were intronic, and their functional exploration in humans remains to be undertaken. However, because we also found that LRIG1 enhanced both BMP signaling and adipogenesis, it seems reasonable to speculate that the diabetes-preventing *LRIG1* alleles may stimulate adipogenesis by enhancing BMP signaling. The diabetes-protecting effect could, thus, result from a more efficient use of excess energy by the LRIG1-mediated increased number of adipocytes[47]. Further analyses of the associations between *LRIG1* gene variants and human metabolic traits may unravel further insights about the physiological function of LRIG1. Nevertheless, LRIG1, LRIG3, or other functionally associated proteins may provide novel targets for the prevention or treatment of type 2 diabetes or other metabolic diseases. However, the molecular mechanisms involved need to be elucidated before potential clinical applications can be explored.

We propose that LRIG proteins play important roles as BMP sensitizers in the context of lipid metabolism, during development, in tissue homeostasis, and in diseases such as cancer. In this regard, it is intriguing that *Lrig3*-deficient mice show both

craniofacial and inner ear defects[48], which are consistent with the central role for balanced BMP signaling during the development of these anatomical structures[49,50]. Similarly, we suggest revisiting the *Lrig1*-deficient phenotypes, including the cutaneous[51,52] and intestinal[9,53] cell hyperproliferation phenotypes, to investigate the role of dysregulated BMP signaling in these processes. In cancer, the roles of LRIG1 and LRIG3 should be re-evaluated in light of their functions as BMP sensitizers. In glioma, for example, LRIG1 functions as a tumor suppressor[11], as does BMP signaling[54–56]. Thus, it can be hypothesized that LRIG1 may suppress glioma growth by enhancing BMP signaling, in addition to its previously proposed regulation of RTK signaling.

Future in vitro cell culture experiments concerning LRIG proteins need to take into consideration the BMP content in FBS. Commercial FBS, which is commonly used for in vitro cell cultivation, contains BMP ligands in concentrations ranging from 6 to 14 ng/ml[57,58], i.e., concentrations where the sensitizing functions of LRIG1 and LRIG3 were highly relevant. Thus, endogenously expressed LRIG proteins are likely to affect BMP signal transduction in cells cultured under standard conditions with FBS.

*Lrig*-null mice are not viable[2]. Nevertheless, the only clear phenotype that we could establish for the *Lrig*-null MEFs was impaired adipogenesis and a reduced sensitivity for BMP4 and BMP6. The *Lrig*-null MEFs did not show any obvious phenotype regarding their viability, morphology, proliferation, migration, energy metabolism, or signaling through TGF-β or RTKs. The lack of a detectable RTK phenotype was particularly intriguing given the substantial body of evidence showing that LRIG proteins regulate RTK signaling. Thus, compared to the wild-type MEFs, the *Lrig*-null MEFs showed no apparent alterations in their steady-state levels of phosphorylated RTKs, PDGF-driven cell proliferation, or PDGF-induced reporter gene activation. The reason for the apparent lack of an RTK phenotype in the *Lrig*-null cells remains enigmatic. Hypothetically, one possible explanation could be a canceling-out effect that occurs when knocking out of genes with opposing functions. For example, LRIG1 and LRIG3 have been shown to have opposing functions with regard to the regulation of ERBB RTKs[59]. It is also possible that the RTK-regulating functions of LRIG proteins become apparent only under specific conditions, such as those observed when signaling proteins are ectopically overexpressed, as has been the case in many of the previous studies[e.g.,5,6,7,8,10,11]. It may also be relevant to consider possible cross-talk between the BMP and RTK signaling pathways, i.e., a primary effect on one of the pathways may indirectly affect the other pathway and vice versa. In this regard, the timing of the different events will be important to resolve to shed light on their causal relationships. Nevertheless, the *Lrig*-null MEFs revealed that the LRIG proteins are not required for basal RTK signaling, at least not in MEFs under standard cell culture conditions.

In summary, we showed that mammalian LRIG proteins function as cellular BMP sensitizers and regulators of adipogenesis. Furthermore, we showed that the *C. elegans* LRIG homolog, *sma-10*, also regulates lipid accumulation in the worm. Importantly, specific human *LRIG1* gene variants were associated with a decreased risk of type 2 diabetes, increased BMI, and altered adipocyte morphology, suggesting that LRIG proteins play important physiological roles in the regulation of lipid homeostasis in humans. It will be important to further investigate the detailed molecular mechanisms involved, which could unravel new molecular players and treatment targets for common human metabolic diseases.

## Methods

**Cell lines and cell culture.** MEFs were isolated from 12-day-old mouse embryos with floxed *Lrig* genes (*Lrig1^{flox/flox}*;*Lrig2^{flox/flox}*;*Lrig3^{flox/flox}*) that had been generated through interbreeding of the previously described mouse strains B6.129-

Lrig1^{tm1Hhed 11}, B6.129-Lrig2^{tm1Hhed60}, and B6.129-Lrig3^{tm1Hhed36} or from embryos with wild-type or deficient *Lrig1* or *Lrig3* genes that had been obtained from inter crosses of B6.129-Lrig1^{tm1.1Hhed11} or B6.129-Lrig3^{tm1.1Hhed} mice[36], respectively, in a C57BL/6 J genetic background. All mice were housed and maintained and all experiments performed in accordance with the European Communities Council Directive (86/609/EEC). Experimental protocols were approved by the Regional Ethics Committee of Umeå University, Umeå, Sweden (registration nos. A5-2010, A193-12, and A1-16). The cells were immortalized according to the 3T3 protocol described by Todaro and Green[61]. The MEFs were cultured in Dulbecco's modified Eagle's medium (DMEM) (Sigma-Aldrich Sweden AB, Stockholm, Sweden) supplemented with 10% FBS (Fisher Scientific GTF AB, Gothenburg, Sweden), MEM-nonessential amino acids (Fisher Scientific GTF AB), 50 μM 2-mercaptoethanol (Sigma-Aldrich Sweden AB), and 50 μg/ml gentamicin (Invitrogen, Fisher Scientific GTF AB). In experiments where MEFs were subjected to multiple washes, the cell culture plates were coated with 0.1% bovine gelatin (Sigma-Aldrich Sweden AB, catalog # G9391) for 30 min at 37 °C prior to cell seeding. To generate *Lrig*-null (*Lrig1^{-/-}*;*Lrig2^{-/-}*;*Lrig3^{-/-}*) cells, herein also referred to as TKO cells, the triple-floxed cells were transduced with adenovirus Ad(RGD)-GFP-iCre or its control adenovirus Ad(RGD)-GFP (Vector Biolabs, Malvern, PA, USA) at a multiplicity of infection of 100 and a cell seeding density of 10,300 cells/cm². Twenty-four hours after adenovirus transduction, the cells were washed with phosphate-buffered saline (PBS), and after an additional 24 h, the top 20% of cells that showed the highest green fluorescence intensity were isolated using a FAC-SAria III cell sorter (BD Biosciences, San Jose, CA, USA). The transduction-selection procedure was repeated independently four times, thereby producing four different cell line pairs composed of an *Lrig*-null and a wild-type MEF line named TKO1-4 and WT1-4, respectively. *LRIG*-inducible MEF lines were generated by stably transducing an *Lrig*-null MEF line with a doxycycline-inducible *LRIG1*, *LRIG2*, *LRIG3*, or empty control vector according to a previously described protocol[11]. The *LRIG1*-inducible human embryonic kidney cell line HEK293T, clone 32:3:10, has been described previously[62], and the human melanoma cell line A375 was obtained from Dr. Oskar Hemmingsson of Umeå University. A375 cells were profiled for short tandem repeats (STRs) by American Type Culture Collection (ATCC) and were confirmed to have a 100% match with the ATCC cell line CRL-1619 (A375). HEK293T cells and A375 cells were cultured in DMEM supplemented with 10% FBS and 50 μg/ml gentamicin. The cell culture plates used for HEK293T and A375 cells were coated with 10 μg/ml poly-D-lysine (Sigma-Aldrich Sweden AB, catalog # P0899) for 30 min at 37 °C followed by washes with PBS before the cells were plated. The *LRIG1*-deficient A375 subclone, clone Pc1-5-4, was generated via CRISPR-Cas9-mediated mutagenesis. To this end, two sgRNAs targeting both strands of *LRIG1* exon 11, were cloned into the *pD1401-AD* plasmid (Atum, Newark, CA, USA), which contains a Cas9(D10A)-GFP-nickase sequence under the CMV promoter. A375 cells were transfected with the resulting plasmid using Lipofectamine 2000 (Fisher Scientific GTF AB) according to the manufacturers protocol. GFP-positive cells were then single-sorted into 96-well plates containing DMEM supplemented with 10% FBS and 200 U/ml penicillin-streptomycin (Fisher Scientific GTF AB) using a BD FACSAria™ III sorter. Single cells were expanded, split, and expanded as duplicates in 6-well plates. One well in each duplicate was lysed and screened for large indels and insertions using PCR (forward primer sequence: 5′-CATTCCATGGGCTTGTGTTG-3′, reverse primer sequence: 5′-CCACTACCATTAATCAGAC-3′). Genomic DNA from a clone that lacked the 278-bp wild-type band was then PCR amplified using primers flanking *LRIG1* exon 11 (forward primer sequence: 5′-GTTTGACTCTAACTCTGTTG-3′, reverse primer sequence: 5′-GCATAATGCAATTGCAGAAG-3′). Each of the three resulting bands were purified and cloned into a TOPO vector (Fisher Scientific GTF AB), sequenced, and found to represent three different mutant variants of *LRIG1*: one with a deletion and one with an insertion, both resulting in frameshifts; the third had a silent intronic insertion, and both PAM sequences were intact. By repeating the entire mutational process on this clone, we isolated the Pc1-5-4 subclone, which was found to contain an additional insertion close to the splice acceptor site at the intron 10/exon 11 boundary. The *LRIG1*-inducible A375 cell line was generated through the cotransduction of the A375 clone Pc1-5-4 with the vectors *pLVX-LRIG1-TRE3G* and *pLVX-Tet3G* as described previously for other cells[11]. Lentiviral particles with vectors for the *srf/elk-1 luciferase* reporter and *Renilla* control were obtained from Qiagen AB (Sollentuna, Sweden, catalog nos. CLS-010L and CLS-RCL, respectively) and were used to cotransduce triple-floxed MEFs with 10 infection units (IU) per cell for *srf/elk-1* and 3.2 IU per cell for *Renilla*. Stably transduced MEFs were selected with puromycin. Thereafter, *Lrig*-null (TKO) and control (WT) MEF lines were generated independently four times from the puromycin-resistant MEFs through transduction of the cells with Ad(RGD)-GFP-iCre or Ad(RGD)-GFP as described above.

**PCR and ddPCR genotype analyses.** The mouse *Lrig* genotypes were routinely monitored via allele-specific PCRs using primers 5′-CATCGCATTGTCTGAG TAGGTGTC-3′ and 5′-CTCCAGAATCACGCTCACCT-3′, yielding an 824 bp product for the floxed wild-type *Lrig1* allele and no product for the knockout allele, primers 5′-TGCACTAGGCAGTCTTAAACCA-3′ and 5′-TCAGGCAGTGACA GAAGGTGT-3′, yielding a 450 bp product for the floxed wild-type *Lrig2* allele and no product for the knockout allele, and primers 5′-CATCGCATTGTCTGAG TAGGTGT-3′ and 5′-CGAGGCTGATGGTCTGCTAAT-3′, yielding a 630 bp

product for the floxed wild-type *Lrig3* allele and no product for the knockout allele. The targeted exon 1 of *Lrig3* together with an untargeted region of *Lrig3* (used as the reference locus) were quantitated using a duplex ddPCR assay. The primers and probes for ddPCR were purchased from Integrated DNA Technologies (Leuven, Belgium). The ddPCR primers used were for *Lrig3* exon 1: 5′-CGCCTTCCCGATC CTCTC-3′ and 5′-GTCTCCTTCACCCCACCG-3′ and for the untargeted *Lrig3* locus: 5′-AACCGTCACCAAGGGAGA-3′ and 5′-CCACCAAAGGGCTGTCATC-3′. The probes used were for *Lrig3* exon 1: FAM-conjugated 5′-ATACTGATACT CACAGCCGTGTGACCCAGG-3′ and for the untargeted *Lrig3* locus, HEX-conjugated 5′-CATTGCTGGAGGGAGCCCGCCC-3′. The final concentrations of forward and reverse primers were 400 nM, and the final concentrations of the probes were 200 nM. DNA, ddPCR Supermix (with no dUTP) (Bio-Rad Laboratories AB, Stockholm, Sweden, catalog # 1863024), Hind III restriction enzyme (Fisher Scientific GTF AB, FastDigest, catalog # FD0505), and nuclease-free water were mixed with primer/probe sets of the targeted and untargeted regions of *Lrig3*. Droplets were generated using a QX200 droplet generator followed by PCR using a T100 thermal cycler (Bio-Rad Laboratories AB) with PCR parameters of 37°C for 5 min, 95 °C for 5 min; 40 cycles of 30 s at 95 °C and 1 min at 58 °C, followed by 98 °C for 10 min. After PCR amplification, the plate was loaded into the QX200 droplet reader (Bio-Rad Laboratories AB) to acquire the data. The data were analyzed using QuantaSoft software (Bio-Rad Laboratories AB, version 1.7.4.0917). Investigators were blinded to the cell line identities at the time of performing ddPCR and data analysis.

**Western blot analysis**. Cells were lysed for 30 min on ice with cell extraction buffer (Invitrogen, Fisher Scientific GTF AB) supplemented with cOmplete, EDTA-free Protease Inhibitor (Roche Diagnostics Scandinavia AB, Bromma, Sweden) and, when analyzing phosphorylated proteins, phosphatase inhibitor PhosSTOP (Roche Diagnostics Scandinavia AB). The lysates were then centrifuged at 20,800 x g for 10 min at 4°C. The resulting pellets were discarded. The protein concentrations of the cleared lysates were determined using a Pierce BCA Protein Assay Kit (Fisher Scientific GTF AB). The protein samples were separated through polyacrylamide gel electrophoresis using 3-8% Tris-acetate gels or 10% Bis-Tris gels (Invitrogen, Fisher Scientific GTF AB, catalog # EA03752 and NP0302, respectively) and then electrotransferred onto polyvinylidene fluoride or nitrocellulose membranes (Bio-Rad Laboratories AB). The membranes were then blocked with Odyssey blocking buffer (LI-COR Biosciences GmbH, Bad Homburg, Germany) or 5% fat-free milk in Tris-buffered saline with 0.1% Tween 20 (TBS-T). The blocked membranes were incubated at 4°C overnight with the primary antibodies at the indicated concentrations (Supplementary Table 10). After three washes with TBS-T, the membranes were incubated with the appropriate secondary antibodies for an hour at room temperature, followed by washes in TBS-T. Thereafter, immune-reactive bands were visualized and analyzed using the Odyssey CLx imaging system (LI-COR Biosciences GmbH) or ECL-select (GE Healthcare, Uppsala, Sweden) together with the ChemiDoc Touch Imaging System (Bio-Rad Laboratories AB). The primary and secondary antibodies used for Western blotting are listed in Supplementary Table 10.

**Cell proliferation assays**. Cell proliferation rates were determined by direct cell counting and an MTT assay. For cell counting, cells were seeded at a density of 2,800 cells per cm² in TC 6-well standard plates (Sarstedt AB, Helsingborg, Sweden). The cells were trypsinized at different times after seeding and counted via the use of a Countess Automated Cell Counter (Invitrogen, Fisher Scientific GTF AB). For the MTT-assay, cells were seeded at the same density in TC 96-well standard plates (Sarstedt AB). Twenty-four hours after the seeding, the medium was changed to cell culture medium containing different FBS or PDGF-BB concentrations. Thereafter, the cells were incubated for an additional 48 h followed by quantification of relative cell numbers via an MTT proliferation kit (Sigma-Aldrich Sweden AB) according to the manufacturer's instructions.

**Cell migration assay**. Migration assays were performed using Corning Transwell cell culture plates with 6.5 mm inserts of 8 μm pore size (Fisher Scientific, GTF AB). Five thousand cells were plated in the upper chamber with medium containing, or not containing, 10% FBS in the bottom chamber. Twenty-four hours after the plating, the membranes were washed with PBS, fixed in ice-cold methanol for 20 min, and stained with 0.1% crystal violet in 20% methanol for 20 min. Five fields from each chamber were counted manually using an Axio Vert.A1 inverted microscope (Carl Zeiss AB, Stockholm, Sweden) equipped with a 5x objective.

**Flow cytometry**. For flow cytometry analyses, cells were dissociated using Accutase cell detachment solution (Sigma-Aldrich, Sweden AB) and then washed in PBS containing 5% FBS. For analysis of intracellular antigens, cells were fixated in 4% phosphate-buffered formaldehyde for 10 min and then permeabilized using 0.2% saponin from Quillaja bark (Sigma-Aldrich, Sweden AB) in PBS for 10 min. The cells were labeled with primary antibodies for 30 min on ice, washed and then incubated with secondary antibodies for 30 min on ice. The primary and secondary antibodies used for flow cytometry analysis are listed in Supplementary Table 10. The flow cytometry analyses were performed on a BD Accuri C6 instrument (BD Biosciences).

**Cell metabolism analyses**. Cell metabolism was analyzed with a Seahorse XFe cell analyzer (Agilent Technologies, Inc., Santa Clara, CA, USA) using the mito stress test and glyco stress test assays according to the manufacturer's instructions. In the mito stress assay, the cells were sequentially treated with 1 μM oligomycin, 1 μM FCCP, and 0.5 μM rotenone and antimycin A. After each treatment, the OCR was measured at three time points. In the glycolytic stress test, the cells were glucose-starved for 1 h followed by sequential treatments with 10 mM glucose, 1 μM oligomycin, and 50 mM 2-deoxy-glucose. After each treatment, the ECAR was measured at three time points. The measurements were normalized to the relative cell numbers, which were determined through the measurement of cell nuclei fluorescence after staining with Hoechst 34580 (Sigma-Aldrich Sweden AB), using a Synergy2 microplate reader (BioTek Instruments SAS, Colmar Cedex, France).

**In vitro adipogenesis assay**. For adipogenic transformation, MEFs were seeded at a density of 28,000 cells/cm² at day −1. At day 0, cells were subjected to an initial adipogenic cocktail containing 1 μM dexamethasone (Sigma-Aldrich Sweden AB), 0.5 mM 3-isobutyl-1-methylxanthine (Sigma-Aldrich, Sweden AB), 10 μg/ml bovine insulin in HEPES buffer (Sigma-Aldrich Sweden AB), and 16 μg/ml rosiglitazone (Sigma-Aldrich, Sweden AB). At day 2, the medium was thereafter changed to a cocktail containing 10 μg/ml insulin and 16 μg/ml rosiglitazone and was changed every two days until day 9 when the cells were fixed with 4% formaldehyde (Unimedic Pharma AB, Stockholm, Sweden) for 30 min and stained using a 60% isopropanol solution with 0.5% Oil Red O (Sigma-Aldrich, Sweden AB). In some experiments, 100 ng/ml recombinant murine noggin (PeproTech Nordic, Stockholm, Sweden, catalog # 250-38) was added at day −1 or 50 ng/ml recombinant human BMP4 (PeproTech Nordic, catalog # 120-05ET) was added at day 0. The Oil Red O stained cells were quantified in a Spectramax i3x plate reader (Molecular Devices, San Jose, CA, USA) using the Softmax Pro 7 software (Molecular Devices).

**RNA-extraction and quantitative RT-PCR-analyses**. For qRT-PCR, RNA was prepared using a PureLink RNA Mini Kit (Invitrogen, Fisher Scientific GTF AB) followed by treatment with PureLink DNase (Invitrogen, Fisher Scientific GTF AB) according to the manufacturer's instructions. The TaqMan gene expression assays for *Lrig1* (Mm00456116_m1), *Lrig3* (Mm00622766_m1), *Cebpb* (mm00843434_s1), *Pparg* (mm00440940_m1), and *Fabp4* (mm00445878_m1) were purchased from Fisher Scientific GTF AB. Primers and probes for *Lrig2* and *RN18S* have been previously described[1]. Data were acquired using a CFX96 system C1000 thermal cycler (Bio-Rad Laboratories AB) as previously described[63]. The specific gene expression levels were normalized to that of *RN18S* by transforming the ΔCT values from log2 to linear values.

**Lipidomics**. One million cells were trypsinized, washed with PBS, and then frozen at −80 °C until use. Lipid extraction and liquid chromatography-quadrupole time-of-flight mass spectrometry-based lipidomics analysis was performed at the Swedish Metabolomics Centre at the Swedish University of Agricultural Sciences (Umeå) as previously described[64].

**Luciferase reporter assays**. Canonical BMP and TGFβ signaling was assessed by transiently transfecting the indicated MEFs with *pGL3-BRE-Luciferase*[65] (Addgene) or *p(CAGA)₁₂MLP-Luc*[42] (kindly provided by Serhiy Souchelnytskyi, Ludwig Institute for Cancer Research, Uppsala, Sweden), respectively. Transfections were performed with Fugene 6 transfection reagent (Promega Biotech AB, Nacka, Sweden) according to the manufacturer's instructions using a 1:3 DNA:reagent ratio, with a DNA amount corresponding to 0.7 μg/cm² and a reporter plasmid:*Renilla* reference plasmid ratio of 1:10. Twenty-four hours after transfection, cells were starved for one hour and treated with BMP4 or recombinant human TGF-β1 (PeproTech Nordic, catalog # 100-21) for three hours. PDGF signaling was assessed using the stably transduced *srf/elk-1 luciferase* MEF lines. Here, the cells were serum-starved for one hour followed by treatment with different concentrations of PDGF-BB (PeproTech Nordic, catalog # 315-18) for four hours. After the treatments with growth factors, cells were lysed using a Dual-Glo Luciferase Assay System (Promega Biotech AB) with a 20-minute incubation time for both assay reagents. Plates were analyzed with a Glomax 96 microplate luminometer (Promega Biotech AB) using an exposure time of 1 s per well. When analyzing transient BMP and TGFβ reporters, the data were normalized by taking the ratio of luciferase/Renilla. For the stably transduced *srf/elk-1* reporter cells, only luciferase was used.

**Phospho-Smad immunofluorescence and Western blot assays**. To analyze BMP- or TGFβ-induced phosphorylation of Smad1/5 and Smad3, respectively, cells were seeded the day before stimulation in a 96-well cell culture microplate (Greiner Bio-One International GmbH, Monroe, NC, USA, catalog # 655090) at densities of 3,000 cells per well for wild-type or *Lrig*-null MEFs, 1,800 cells per well for *LRIG*-inducible MEFs, and 10,000 cells per well for HEK293T and A375 cells. LRIG expression was induced in *LRIG*-inducible cells by treatment of the cells with 100 ng/ml, unless otherwise indicated, of doxycycline (Clontech Laboratories, Bio-Nordika Sweden AB, Stockholm, Sweden) for 24 hours prior to starvation. Cells were starved in serum-free cell culture medium for one hour and then stimulated with BMP4, recombinant human BMP6 (PeproTech Nordic, catalog # 120-06), recombinant human GDF2/BMP9 (PeproTech Nordic, catalog # 120-07), or TGF-

β1 for one hour. Thereafter, the cells were fixed with 4% formaldehyde for 10 min, permeabilized with 0.2% saponin for 10 min, and blocked in blocking buffer composed of PBS, 0.1% Tween 20, and 5% FBS. After blocking, cells were incubated overnight with the appropriate primary antibody followed by washes and an incubation for one hour with the corresponding secondary fluorescent antibodies. For cell number normalization, 1 μg/ml Hoechst 33342 was added before analysis. Plates were imaged using a whole-well imaging device Trophos Plate Runner HD (Trophos/Dioscure, Marseille, France) with exposure time set to detect stained area. Images were analyzed using the Tina analysis package (Trophos/Dioscure) adjusting the threshold to remove background and rolling ball subtraction. Objects at specified sizes were detected, and their fluorescence was normalized using Hoechst 33342 nuclear staining by dividing the number of cells (for MEFs and A375 cells) or the mean cell fluorescence (for HEK293T cells). For Western blot analysis, cells were seeded into 6-well plates at a density of 10,344 cells/cm$^2$. Two days after seeding, cells were starved for one hour in serum-free medium and then stimulated with 5 or 20 ng/ml BMP4 or serum-free medium for one hour before lysis. The antibodies used for the phospho-Smad immunofluorescence and Western blot analyses are listed in Supplementary Table 10.

**Transcriptomics**. Transcriptomes were analyzed via RNAseq. To this end, 500,000 cells were serum-starved for 1 h prior to cell lysis. RNA was isolated using the Dynabeads mRNA DIRECT Purification Kit (Fisher Scientific GTF AB) according to the manufacturer's instructions. The purity and integrity of the RNA preparations were confirmed with an RNA 6000 Nano kit and an Agilent Bioanalyzer (Agilent Technologies). Sequencing was performed at SciLifeLab (Uppsala, Sweden) with an Ion Technology sequencer Ion Proton (Fisher Scientific GTF AB). Reads were aligned using STAR and bowtie2 software, and HTSeq was used to generate counts.

**Correlations between LRIG1-GFP variants and BMP-induced phosphorylation of Smad1/5**. Cells were transiently transfected using Fugene 6 as described above. *pLRIG1-GFP* encoding full-length LRIG1 fused to GFP has been described, previously[66]. *pLRIG1-Δcyto-GFP* encoding the extracellular/luminal and transmembrane parts, together with the first three cytosolic amino acids (YQT), of LRIG1 fused to GFP was generated by cloning the corresponding PCR-amplified *LRIG1* fragment into the *pEGFP-N1* (Clontech Laboratories) expression vector. PCR was used to generate *pLRIG1-3XFLAG* and *pLRIG1-ecto-3XFLAG* by amplifying the regions corresponding to the full length and the ectodomain of LRIG1, respectively, from an *LRIG1* cDNA (GenBank accession no. AF381545) and cloning these fragments into *p3XFLAG-CMV-13* (Sigma-Aldrich Sweden AB). The integrity of *pLRIG1-3XFLAG* and *pLRIG1-ecto-3XFLAG* were confirmed by DNA sequencing using a Big Dye Terminator v 3.1 cycle sequencing kit (Fisher Scientific GTF AB) and a 3730xl DNA analyzer (Fisher Scientific GTF AB). Twenty-four hours after transfection, Smad1/5 phosphorylation immunofluorescence assays were performed using 20 ng/ml BMP4 for 20 min. Both GFP fluorescence and pSmad1/5 immunofluorescence were quantified simultaneously using a Trophos plate runner.

**RTK array**. To compare the RTK phosphorylation levels, whole-cell lysates containing 150 μg protein were analyzed using a Human Phospho-RTK Array Kit (R&D Systems Europe Ltd., Abingdon, UK; catalog # ARY014) according to the manufacturer's instructions and quantified using ChemiDoc Touch Imaging System and Image lab software (Bio-Rad Laboratories AB).

***C. elegans* analyses**. All *C. elegans* strains used are described in WormBase (www.wormbase.org). N2 Bristol was used as the wild-type strain in all cases. Worms were maintained at 20 °C on standard nematode growth medium (NGM) agar and with *E. coli* OP50 as a food source. One-day-old adult worms from staged plates were stained with Oil Red O as described previously[67] and imaged at a midplane, with the mouth and the pharyngeal lumen in focus, using the 10x objective on a DIC-equipped Olympus BX51 microscope. Color images were then subtracted for background (rolling ball radius: 50.0 pixels) and converted to 8-bit CIELAB using Fiji software[68]. Quantifications were performed in the "a" channel by selecting an area averaging ~2,500 μm$^2$ between the posterior end of the pharynx and the anterior border of the gonad to avoid the signal coming from the oocytes. The local background signal for each measurement was then subtracted. All genotypes under study were analyzed in parallel, and each experimental round was normalized to the combined mean signal, which was calculated from all genotype means.

**Statistics and reproducibility of cell and animal experiments**. All Student's *t*-tests were 2-sided. All statistical analyses of cell and animal experiments were performed using GraphPad Prism 8 software (La Jolla, CA, USA), and the *P*-values <0.05 were considered significant. When cell lines were compared, in general, data were obtained from at least four biological replicates per genotype and three experimental repeats performed on separate days. The exact number of replicates are presented in the individual figure legends.

**UK Biobank population characteristics**. The UK Biobank is a large project with genotyped and well-phenotyped individuals comprising approximately 500,000

participants[69]. In this study, we excluded participants who did not have BMI measurements or of any of the other outcome variables of interest (done at the stage of that particular analysis) and those who had ambiguous information on sex (discordance between self-reported and genetically encoded sex). The final sample size was 398,810 participants of Caucasian ancestry. Supplementary Table 3 shows the participants' characteristics, and Supplementary Table 4 shows the *LRIG1* tag SNP information. This study was conducted using publicly available data from the UK Biobank, and therefore the current analyses did not require specific ethical approval. The reference for the approved UK Biobank project is ukb18274.

**Tag SNP identification**. We used *snptag*, an online tool of SNPinfo, (https://snpinfo.niehs.nih.gov/snpinfo/snptag.html) to identify tag SNPs from the nine *LRIG1* SNPs that were significant in the genome-wide association (GWA) analysis. In our tag SNP selection, the population was restricted to the Utah residents with Northern and Western European ancestry from the CEPH collection (CEU); the linkage disequilibrium (LD) threshold was set at r2 = 0.8, the maximum distance (bp) between SNPs for calculating the LD was 250,000 bp, and the minor allele frequency (MAF) range was 0.01 to 0.5. Three tag SNPs and one non-synonymous SNP were identified from the search (Supplementary Fig. 6).

**Association with type 2 diabetes and BMI**. To isolate the effect of BMI attributed to genetic variance, we regressed out the effects of age, age squared, sex, batch effect, and the first ten genetic principal components separately for men and women and extracted the BMI residuals. The residuals were then used in subsequent analyses as (a) untransformed residuals, and, (b) as inverse-normal transformed residuals. Each case is indicated in the respective analyses. We used these transformed residuals to create interaction terms with each of the tag SNPs and investigated the associations between these interaction terms and the risk of type 2 diabetes using logistic regression models adjusted for age, batch effect (array type used for genotyping, UK BiLEVE or UKBB Axiom) and the first ten genetic principal components. Type 2 diabetes was diagnosed by a doctor in this population and was parameterized as a binary outcome, coded "Yes" or "No", for these analyses. We also tested the association of each of the SNPs with type 2 diabetes using logistic regression models adjusted for untransformed BMI residuals and another model without BMI, in addition to the covariates mentioned above in the first model. We investigated the relationship between BMI and each of the SNPs using a simple linear regression model with untransformed BMI residuals as the outcome. All analyses were based on an additive genetic model.

**Association with liver fat and biochemical measures of adiposity**. To investigate the relationship between these tag SNPs and liver fat percentage (LF%) among 3,858 UK Biobank participants who had liver fat measurements, we extracted LF% residuals separately in men and women using multiple linear regression models adjusted for age, age squared, sex, BMI, array batch, and the first ten genetic principal components. The residuals were then inverse-normal transformed, and a simple linear regression model was fitted to test the association between each of the SNPs with LF% residuals as an outcome. For each of the two biochemical measures (baseline total cholesterol (mmol/L) and triglyceride levels (mmol/L)), we extracted residuals for participants with the relevant measures as outlined above. Cholesterol was normally distributed, so the residuals were not further transformed, but those for triglycerides were inversely normal transformed. The association between each of the measures and each of the tag SNPs was modeled using a simple linear regression model with the respective residuals as the outcome. In all analyses, the minor allele was used as the reference allele.

**GENiAL study participants**. The GENetics of Adipocyte Lipolysis (GENiAL) cohort included 273 men and 718 women and have been described previously[70]. Briefly, subjects in the GENiAL cohort were recruited by local advertisement to examine the regulation of fat cell function. Fifty-seven percent of the participants were obese (defined as BMI ≥ 30 kg/m$^2$). They all lived in Stockholm County, Sweden and were at least second-generation Swedes. One hundred ninety-four participants had type 2 diabetes, hypertension, or dyslipidemia alone or in different combinations. None were treated with insulin, glitazones, or glucagon-like-peptide analogs. Data on clinical variables are summarized elsewhere[70]. The study was approved by the local ethics committee at the Huddinge University Hospital (D. no. 167/02, 2002-06-03) and was explained in detail to each participant. Informed consent was obtained from all participants. Included in this study were 948 subjects from the GENiAL cohort with adipose morphology data available[70].

**Clinical examination**. The GENiAL participants came to the hospital's clinical research center the morning after an overnight fast. Their heights, weights, and waist-to-hip ratios (WHRs) were measured. Each participant's body fat content was measured by bioimpedance, and their total body fat mass was indirectly calculated using a formula based on age, sex and BMI[71]. A venous blood sample was obtained for extraction of DNA and clinical chemistry by the hospital's accredited routine clinical chemistry laboratory. Subcutaneous adipose tissue (SAT) was obtained by a needle aspiration biopsy lateral to the umbilicus as previously described[72].

**Adipose tissue phenotyping**. SAT samples were rapidly rinsed in sodium chloride (9 mg/ml) before removal of visual blood vessels and cell debris and were subsequently subjected to collagenase treatment to obtain isolated adipocytes as described[73]. The mean weight and volume of remaining cells were determined as described[74]. A curve fit of the relationship between mean adipocyte volume and estimated abdominal subcutaneous fat mass was performed[75]. The difference between the measured and expected mean adipocyte volume at the corresponding total fat mass determines adipose morphology. If the measured adipocyte volume is larger than expected, SAT hypertrophy prevails, whereas the opposite is true for hyperplasia. Thus, this measure of adipose morphology is independent of total fat mass.

**Genetic analysis of the GENiAL cohort**. The genetic analysis of the GENiAL cohort has been described previously[70]. After quality control, 894 samples were available for analysis. Genetic association analysis was conducted in PLINK[76], using linear regression, assuming an additive genetic model, and adjusting for population structure (PCs1-3), age, and sex.

**Reporting summary**. Further information on research design is available in the Nature Research Reporting Summary linked to this article.

## Data availability

The transcriptomic RNAseq datasets generated and analyzed during the current study are available in the NCBI BioProject repository, https://www.ncbi.nlm.nih.gov/bioproject/PRJNA684140. Relevant data and/or materials not present are available upon reasonable request from C.He. (carl.herdenberg@umu.se) and/or H.H. (hakan.hedman@umu.se). Source data from the UK Biobank are available upon approved application through the UKBB data application management system. Source data for the main figures are presented in Supplementary Data 1.

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

## Acknowledgements

We thank Jonas Nilsson (Umeå University) for constructing the *pLRIG1-Δcyto-GFP* vector, Mahmood Faraz (Umeå University) for constructing the inducible *LRIG* vectors, Bhavna Rani (Umeå University) for assembling the *pLRIG1-3XFLAG* and *pLRIG1-ecto-3XFLAG* vectors, Richard Padgett (Rutgers University, NJ, USA) for nematode strains, Charlotte Nordström, Yvonne Jonsson, and Annika Holmberg for technical assistance, Mehmet Dinçer Inan for help with Oil Red O staining of *C. elegans* worms, Jonathan Gilthorpe (Umeå University) for valuable input regarding the Trophos and Seahorse analyses, the Swedish Metabolomics Centre (Umeå University) for help with lipid analysis, Neli Tsereteli (Lund University Diabetes Center) for assistance with data visualization, and Alaitz Poveda (Lund University Diabetes Center) for help with data preparation. We also thank all participants and staff involved in the GENiAL study. This work was supported by grants from the Swedish Cancer Society (CAN 2016/344 and CAN 2018/546), Kempe Foundation (JCK-1829), the Cancer Research Foundation in Northern Sweden (AMP 16-799), Lion's Cancer Research Foundation at Umeå University (LP 16-2134 and LP 15-2057), the Strategic Research Program in Diabetes at Karolinska Institutet, the Swedish Research Council, Novo Nordisk Foundation, the Swedish Diabetes Association, and by the regional agreement between Umeå University and Västerbotten County Council on the cooperation in the field of Medicine, Odontology and Health (ALF, RV 836951). The work undertaken by PWF, PM, and AP was supported by the European Research Council (CoG-2015_681742_NASCENT), the Swedish Research Council (Distinguished Young Researchers Award in Medicine), the Swedish Heart-Lung Foundation, and the Novo Nordisk Foundation. RJS was supported by a UKRI Innovation at HDR-UK Fellowship (MR/S003061/1).

## Author contributions

C.He. and H.H. conceived the idea, planned the experiments, and wrote the manuscript. H.H. and R.H. supervised the project. C.He. developed the MEF lines, pSmad immunoassay, and adipogenic assay, and performed the in vitro experiments. O.B. and S.T. investigated the connection between *sma-10* and fat accumulation in *C. elegans*. C.Ho. contributed to the initial MEF triple KO experiments. A.A. investigated noncanonical BMP signaling and PDGF-driven cell proliferation and performed the BMP assays (Figs. 2c, d and 5d). P.M.M. and P.W.F. investigated the associations between *LRIG1* gene variants and BMI and diabetes. R.J.S., I.D., and P.A. investigated the correlation between *LRIG1* gene variants and adipose tissue morphology. All authors reviewed the final draft.

## Funding

## Competing interests

The authors declare no competing interests.
