## [Peer Review File · Communications Biology]

Reviewers' comments:

Reviewer #1 (Remarks to the Author):

In this manuscript, Herdenberg et al. convincingly show that LRIG proteins mediate BMP signaling to regulate lipid metabolism. The work is thorough and clearly presented. A strength of the paper is that the investigators demonstrate concordant results using three different systems: mouse cells, *C. elegans*, and human subjects. The conclusions are inherently important due to the implications for understanding lipid metabolism, but also because they address controversy in the field surrounding the signaling pathways with which LRIG proteins interact.

Minor comments:

Figure 1A summarizes a large amount of data that should be more readily accessible to the reader. First, the authors indicate that there is a large amount of variability in the results, so it is important to plot the data for individual trials rather than only the combined column graph. Second, it would be helpful to show images of the stained cells which would provide added insight beyond a simple % area stained calculation. The morphology of the cells is discussed later in the context of the human experiments, but there is no reference to cell size/number/proportion of adipocytes in this experiment.

In Figure 4C, the induction of pp38 by BMP4 in LRIG1 expressing cells is not very convincing. A positive control is needed to demonstrate that this fold increase is relevant.

Heading of last section of results, "Human LRIG1 variants are associated with altered risk of type 2 diabetes, BMI, and adipocyte morphology" could be reworded. There is not a risk of BMI and adipocyte morphology, but of higher or lower BMI and abnormal or altered morphology.

In Figure 7A, the authors see no significant change in lipid signal in *sma-6* mutants, which differs from published observations. This discrepancy should be discussed.

Figure 7D labeling is not consistent with the text. Choose one nomenclature for the SNP for clarity (either A/C or T/G). It would also be helpful to label the common and rare genotypes in the figure.

What is the amino acid change associated with this LRIG SNP? Can the authors speculate on how it might alter the protein function?

Reviewer #2 (Remarks to the Author):

In the manuscript Håkan Hedman and colleagues show leucine-rich repeats and immunoglobulin-like domains (LRIG) proteins have roles in regulating adipogenesis and BMP signaling. The author used *Lrig*-null mouse embryonic fibroblasts (MEFs) and *C. elegans* with mutant LRIG/*sma-10* variants to explore the roles of LRIGs in adipogenesis and lipid metabolism. They also analyzed human LRIG1 variants and identified the alleles that were strongly associated with increased body mass index (BMI). The previously unidentified function of LRIGs in lipid metabolism and BMP signaling may have implications for human disease. There are some concerns.

The authors concluded that *Lrig*-null MEFs had impaired ability of adipocyte differentiation by comparing Oil Red O stained adipocytes after adipogenic induction. The authors described that three out of four wild-type MEF lines were able to transform into adipocytes in response to the adipogenic cocktail, whereas none of the *Lrig*-null MEF lines did that. However, although the Oil red O stained area of the wild type cells was only about 6%, *Lrig*-null cells still got the value of about 2%. So it is not proper to say WT cells can transform into adipocytes, while *Lrig*-null cells cannot.

What's the purpose of the lipid chromatography? Regarding there were cells have differentiated into mature adipocytes in both groups of MEFs, did the experiments aim to show the defect of adipogenic differentiation, or changes lipid profiling in mature adipocytes? Did the authors pooled all the differentiated and undifferentiated cell for lipid chromatography? Was the deviation showed by the chromatography resulted from the variation of percentage of differentiated cells, or the changes of mature adipocyte per se?

It seems that there still some Lrig-null cells had differentiated into adipocytes. As the authors also showed the variation of lipid composition between the two groups of cells. Does the null of LRIGs affect lipid metabolism of mature adipocytes derived from MEFs? The adipogenic rate of the MEF cells seems to be very low even in the wild type cells. Would the low rate of differentiation bias the actual changes showing up in post differentiation stage? In other words, the actual changes in the mature adipocytes between two groups was interfered by the difference between precursors and mature adipocytes.

C. elegans with mutant LRIG/sma-10 variants showed abnormal lipid deposits. It seems that LRIGs regulate the lipid metabolism, so when reviewing the phenotype of Lrig-null MEFs again, it raises the question that whether the decreased coverage of Oil red O staining was due to less amount of lipid in adipocytes rather than less number of differentiated adipocytes.

What about the RTK signaling level in C. elegans with mutant LRIG/sma-10 variants?

Is there any reports about model organisms of LRIGs deficient specifically in adipose tissue?

How about expression of LRIGs before and after differentiation? Which stage does LRIG-BMP function in? Pre differentiation, during differentiation, or post differentiation? Have the authors ever compared the levels of LRIG1-3 in undifferentiated and differentiated MEFs, as well as the relative levels of the three types of protein?

LRG null desensitized MEFs to response the BMPs, moreover there are deviation among bmp4, 6, and 9. It seems that it's not the receptors that lead to the difference as the authors have examined the level of different types of receptors. What's the possible target of LRIGs on BMP signaling? Do the LRIGs affect levels of BMPs? What's the possible reason for that maximal pSmad1/5 response did not appear to differ between the wild-type and Lrig-null MEFs (Fig.2b).

Both smad and Mitogen-activated protein kinases/extracellular signal-regulated kinase (MAPK/ERK) pathway is supposed to be associated with the cell proliferation. Why the alterations of smad and MAPK in Lrig-null MEFs had no effect on cell numbers? To our knowledge, BMP4 can stimulate proliferation of C3H10t1/2, which was also derived from mouse embryonic fibroblasts. Regarding the Lrig-null change the sensitivity of cell responding to BMP4, would the author determined the proliferation rate after treating cells with different concentration of BMP4? Why Lrig-null MEFs in culture with about 5%FBS got higher OD value in MTT assay in Fig. S1d.

Some alleles of Lrig1 associated with increased the BMI and hyperplasia of adipocytes? How do they affect LRIG1? Gain of function or loss of function?

Minor comment :

In Figure S1f,g, the authors examined the aerobic and anaerobic metabolism of WT and Lrig-null MEFs, what about OCR and ECAR in differentiated adipocytes derived those MEFs?

In figure 4, why the dox is 100ng/ml in A-D, while 500ng/ml in E-H?

In figure 5A pSmad1/5 was analyzed following the cell were stimulated with 2.5ng/ml BMP4 for one hour, however in Figure 5b, c the concentration of BMP4 was 20ng/ml and treatment lasted 20

minutes. Why the concentration and duration were different? If BMP4 in Figure 5b, c is the same as in Figure 5A, will it be different?

In figure 7c, how was the Oil red O signal normalized? Is it divided by body length?

Reviewer #4 (Remarks to the Author):

The manuscript "LRIG proteins regulate lipid metabolism via BMP signaling and affect the risk of type 2 diabetes" by the group of Dr. Hedman provides evidence that mammalian LRIG proteins function as cellular BMP sensitizers and their ablation impairs adipogenesis in mouse embryonic fibroblasts. Lipid storage defects have also been shown in *C. elegans* with mutant LRIG/sma-10. More importantly, they showed a strong association between specific human LRIG1 gene variants with a decreased risk of type 2 diabetes, increased BMI, and altered adipocyte morphology, suggesting that LRIG proteins play important physiological roles in the regulation of lipid homeostasis in humans.

Overall this is a very well written and interesting study which is of high potential interest for scientists in this field, but a number of issues should be clarified to fully understand the observed findings.

1) Western blot analysis for LRIGs during different stages of adipogenesis in wild-type MEF lines would give an important information w.r.t adipocytic transformation.

2) Did authors observe any difference in transcript levels of BMP4 (or other BMPs) and genes associated with BMP-signaling pathway (such as SMADs, ID genes etc) between the wild-type and Lrig-null MEF lines?

3) In Figure 2G, the blots for SMAD and P38 should be included (as they did for Figure 4E) and used for the respective calculations.

Also, authors showed through western blot analysis of pSMAD1/5 (Figure 4E) that ectopic LRIG1 rescued BMP signaling (upon stimulation with 20ng/ml BMP4), but no band was observed in -dox (empty vector/LRIG) or +DOX (empty vector) (or one can say in "Lrig-null MEFs") with 20 ng/ml BMP4 stimulation. Whereas a strong band for pSMAD1/5 was observed upon BMP4 stimulation (20ng/ml) in Lrig-null MEFs in Figure 2G. Please clarify.

4) Authors are requested to show individual values (i.e. interleaved scatter plot) for the bar graphs in Figure 1 & 2.

5) Using Immunofluorescence and western blot analysis of pSMAD/p38 (figure 4), authors demonstrates that ectopic LRIG1 and LRIG3 rescued BMP, but authors should also study the effects on lipid accumulation or markers of adipogenesis to see whether it restores adipogenesis in Lrig-null MEF lines or not?

Reviewer #1 (Remarks to the Author):

In this manuscript, Herdenberg et al. convincingly show that LRIG proteins mediate BMP signaling to regulate lipid metabolism. The work is thorough and clearly presented. A strength of the paper is that the investigators demonstrate concordant results using three different systems: mouse cells, C. elegans, and human subjects. The conclusions are inherently important due to the implications for understanding lipid metabolism, but also because they address controversy in the field surrounding the signaling pathways with which LRIG proteins interact.

Minor comments:

Figure 1A summarizes a large amount of data that should be more readily accessible to the reader. First, the authors indicate that there is a large amount of variability in the results, so it is important to plot the data for individual trials rather than only the combined column graph. Second, it would be helpful to show images of the stained cells which would provide added insight beyond a simple % area stained calculation. The morphology of the cells is discussed later in the context of the human experiments, but there is no reference to cell size/number/proportion of adipocytes in this experiment.

A1: In the revised manuscript, we have changed former Figure 1A-D, H, and I to show individual values and the means instead of the previous bar graphs (Figure 1C, G-I, M, and N in the revised manuscript). We have also added some representative micrographs showing the Oil Red O stained cells (Figure 1A and B in the revised manuscript).

In Figure 4C, the induction of pp38 by BMP4 in LRIG1 expressing cells is not very convincing. A positive control is needed to demonstrate that this fold increase is relevant.

A2: We acknowledge that the non-canonical induction of pp38 by BMP4 may not seem very convincing. However, we observed that BMP4 induced only a 40% increase in pp38 in our experimental system using wild-type MEFs (Figure 2I). Nevertheless, we found that this (modest) increase in pp38 was Lrig-dependent (Figure 2G, I) and could be rescued by ectopic LRIG1 expression (Figure 3G, $p < 0.001$), but not by LRIG3 expression (Figure 3H). In the revised manuscript, we have further investigated the relevance of this finding by extending our analysis to include also the other MAPK-family members, Erk1/2 and Jnk. This analysis showed that non-canonical activation of Jnk was also LRIG-dependent and could be rescued by LRIG1 (Figure 3I), but not by LRIG3 (Figure 3J), whereas pErk1/2 was not induced by BMP4 under the experimental conditions used (Figure 3K, L). Regarding the biological relevance of non-canonical BMP signaling, this question was not addressed in the present study. For this, we refer to the available literature, including references (21,22) in the manuscript.

Heading of last section of results, “Human LRIG1 variants are associated with altered risk of type 2 diabetes, BMI, and adipocyte morphology” could be reworded. There is not a risk of BMI and adipocyte morphology, but of higher or lower BMI and abnormal or altered morphology.

A3: We have reworded the heading of the last section, to read: “Human LRIG1 variants are associated with an altered risk of type 2 diabetes and with BMI and adipocyte morphology”,

page 12, lines 283-284.

In Figure 7A, the authors see no significant change in lipid signal in sma-6 mutants, which differs from published observations. This discrepancy should be discussed.

A4: Using Raman spectroscopy, Yu et al. showed that *sma-6(wk7)* mutants have significantly lower somatic fat levels than wild-type animals (Yu et al., 2017). Our Oil Red O results showed the same trend, with 33% lower fat levels in the *sma-6(wk7)* mutants than the wild-type worms, but the difference was not statistically significant ($p=0.14$). The lack of significance in our *sma-6(wk7)* results might be attributed to the relatively high variance observed for the measurements of this worm series. The reason for the high Oil Red O variance among the *sma-6(wk7)* worms is not known; however, more Oil Red O measurements would likely resolve whether *sma-6(wk7)* fat levels differ from wild-type. However, we felt that this was beyond the scope of the present study, so we did not pursue this line of investigation. To clarify the statistical problem associated with the high variance among *sma-6(wk7)* worms, the Results have been re-written accordingly (page 12, lines 275-277).

Figure 7D labeling is not consistent with the text. Choose one nomenclature for the SNP for clarity (either A/C or T/G). It would also be helpful to label the common and rare genotypes in the figure.

A5: In the text, two SNPs are mentioned, rs4856886 and rs9840088, in two different analyses. Rs4856886 is mentioned in relation to population level analysis and rs9840088 in regards to adipocyte morphology analyses. The rare/minor genotype for rs4856886 is G, while the common one is T. In Figure 6D, we present the population level associations of the rs4856886 SNP with BMI and T2D, with the minor/major alleles indicated in the figure caption. The minor allele had not been referenced in the text but this has now been done for clarity (p.13, line 290): “In our analyses, each copy of the minor allele of rs4856886 (G) (tag SNP with the strongest effect) increased the BMI by $\sim 0.05 \text{ kg/m}^2$; this phenomenon was largely attributed to genetic variation (Table S5).”

In the text, the SNP rs9840088 is referenced in relation to adipocyte morphology analyses (another level of analysis for this study) and the common/minor alleles indicated correctly as A/C. See p. 13, lines 305-307: “For rs9840088, the mean adipose morphology values were +7.5 picolitres for the common A allele and -9.2 picolitres for the minor C allele ($P=0.026$ by analysis of covariance, adjusted for age).”

Caption for the updated plot (p. 50, lines 1222-1239): Figure 6D. Plot illustrating the difference in predicted BMI (least square means, LSMs) across the genotypes of rs4856886 (minor allele = G, major allele = T) and odds ratio for type 2 diabetes (T2D) across genotypes. The x-axis represents the rs4856886 genotypes compared. The y-axis on the left represents the difference in BMI LSMs per single minor allele across the genotypes, while the y-axis on the right represents the odds ratios for T2D. In this study, the minor allele was associated with an increase in BMI and a lower odds of T2D risk

What is the amino acid change associated with this LRIG SNP? Can the authors speculate on how it might alter the protein function?

A6: This SNP is an intronic variant, meaning that its mRNA transcripts, together with the intragenic region they belong to, are spliced during pre-translation processing. As mentioned in the text, functional exploration in humans has not yet been undertaken, which therefore presents an interesting avenue for further studies (p. 14-15, lines 364-365). However, to provide a brief background for clarification, from a GWAS perspective, genotyped SNPs may not be necessarily causal but they usually tag the causal variant within the same haplotype block. As such, this SNP tags a protein-coding gene, *LRIG1*, which was shown herein to regulate BMP signaling. According to the ensemble variant effect predictor (https://useast.ensembl.org/Homo_sapiens/Tools/VEP), which collates genetic variant transcription information from multiple sources, this SNP is a modifier (including the other mentioned SNP, rs9840088), and no amino acid changes have yet been identified, further reinforcing the need for additional functional studies.

Reviewer #2 (Remarks to the Author):

In the manuscript Håkan Hedman and colleagues show leucine-rich repeats and immunoglobulin-like domains (LRIG) proteins have roles in regulating adipogenesis and BMP signaling. The author used Lrig-null mouse embryonic fibroblasts (MEFs) and C. elegans with mutant LRIG/sma-10 variants to explore the roles of LRIGs in adipogenesis and lipid metabolism. They also analyzed human LRIG1 variants and identified the alleles that were strongly associated with increased body mass index (BMI). The previously unidentified function of LRIGs in lipid metabolism and BMP signaling may have implications for human disease. There are some concerns.

The authors concluded that Lrig-null MEFs had impaired ability of adipocyte differentiation by comparing Oil Red O stained adipocytes after adipogenic induction. The authors described that three out of four wild-type MEF lines were able to transform into adipocytes in response to the adipogenic cocktail, whereas none of the Lrig-null MEF lines did that. However, although the Oil red O stained area of the wild type cells was only about 6%, Lrig-null cells still got the value of about 2%. So it is not proper to say WT cells can transform into adipocytes, while Lrig-null cells cannot.

A7: We apologize for this inaccurate statement. However, it should be noted that the Oil Red O method used for quantification presented a relatively high background noise; even areas devoid of apparent adipocytes scored between 0.5-1% positive. Nevertheless, it is true that some adipocytes also developed among the *Lrig*-null MEFs, and BMP4 could clearly rescue the apparent deficiency of the *Lrig*-null cells. Thus, to accommodate the findings that not all wild-type cells underwent adipogenesis and that some *Lrig*-null cells did, the title of the corresponding Results section has been changed from “*Lrig* proteins are required for adipogenesis in vitro” to “*Lrig* proteins promote adipogenesis in vitro” (p. 6, line 126), and the running text has also been modified accordingly (p. 6, lines 130-133).

What's the purpose of the lipid chromatography? Regarding there were cells have differentiated into mature adipocytes in both groups of MEFs, did the experiments aim to show the defect of adipogenic differentiation, or changes lipid profiling in mature adipocytes? Did the authors pooled all the differentiated and undifferentiated cell for lipid chromatography? Was the deviation showed by the chromatography resulted from the variation of percentage of differentiated cells, or the changes of mature adipocyte per se?

A8: The purpose of the lipid analyses was to confirm and biochemically extend the Oil Red O findings. This purpose is stated on page 7, lines 145-148. We have also added a new section to the Discussion (page 13, lines 318-331) where we discuss different interpretations of the adipogenesis results.

The lipid analyses showed no apparent difference in the lipid composition between the genotypes in the undifferentiated cells. This finding is stated on page 7, lines 151-153. Additionally, yes, the cells were pooled, which we hope is now clear in the reworded Results sentence (page 7, lines 148).

Regarding the question of whether the observed differences in lipid profiles resulted from the variation in percentage of differentiated cells or the changes in mature adipocyte per se, our experiments were not designed to directly address this question. However, using an operational definition of adipocytes based on morphological criteria, Oil Red O staining, and the previously recognized adipogenic markers Pparg and Ap2, we demonstrated that the phenotypic difference could be classified as adipogenic differentiation rather than altered lipid metabolism in differentiated adipocytes. A section has been added in the Discussion to discuss this issue further (page 14, lines 318-331).

It seems that there still some Lrig-null cells had differentiated into adipocytes. As the authors also showed the variation of lipid composition between the two groups of cells. Does the null of LRIGs affect lipid metabolism of mature adipocytes derived from MEFs? The adipogenic rate of the MEF cells seems to be very low even in the wild type cells. Would the low rate of differentiation bias the actual changes showing up in post differentiation stage? In other words, the actual changes in the mature adipocytes between two groups was interfered by the difference between precursors and mature adipocytes.

A9: Reviewer #2 brings up an important question, i.e., whether LRIG proteins regulate adipogenesis per se and/or the metabolism of specific cell types. The reviewer also notes that the adipogenesis efficiencies among the MEFs were relatively low. The low adipogenesis efficiency is probably due to our use of primary MEF lines rather than specifically selected pre-adipocytic MEF lines, such as 3T3-L1, which are often used in adipogenesis research. Nevertheless, the added images (Figure 1A, B) show that the size of the differentiated adipocytes was highly uniform, and the quantification method was based on the covered area and not the intensity of staining, thus, showing a difference in numbers of adipocytes per se. The interpretation that Lrig proteins regulate adipogenesis per se is also supported by the observation that the induction of early adipogenic markers differed between wild-type and Lrig-null cells at day 4, even before any visible adipocytic transformation had occurred. We have added a section to discuss some of these issues in the Discussion section (page 14, lines 318-331).

C. elegans with mutant LRIG/sma-10 variants showed abnormal lipid deposits. It seems that LRIGs regulate the lipid metabolism, so when reviewing the phenotype of Lrig-null MEFs again, it raises the question that whether the decreased coverage of Oil red O staining was due to less amount of lipid in adipocytes rather than less number of differentiated adipocytes.

A10: Again, we agree that this is an intriguing question. However, based on the gene and protein expression results shown in Figure 1H, I, demonstrating that the induction of Pparg and Ap2 was impaired in the Lrig-null MEFs, we inferred that adipocyte differentiation per se was affected in these cells.

What about the RTK signaling level in C. elegans with mutant LRIG/sma-10 variants?

A11: Using the vulval induction model in *C. elegans*, we obtained data showing that SMA-10 could regulate Ras signaling in the presence of the receptor tyrosine kinase EGFR/LET-23. These results will be published in a separate manuscript. LRIG research has until now been highly divided between studies on BMP signaling in *C. elegans* and RTK signaling in vertebrates. In the current manuscript, we have taken the first step toward bridging this divide by showing for the first time that LRIG proteins also regulate BMP signaling in vertebrates. We look forward to seeing what future comparative research will reveal about the functions in LRIGs.

Is there any reports about model organisms of LRIGs deficient specifically in adipose tissue?

A12: No, to our knowledge there is no such report. Therefore, we have applied for grants to knock out *Lrig* genes specifically in adipose tissue, liver, or skeletal muscle in mice.

How about expression of LRIGs before and after differentiation? Which stage does LRIG-BMP function in? Pre differentiation, during differentiation, or post differentiation? Have the authors ever compared the levels of LRIG1-3 in undifferentiated and differentiated MEFs, as well as the relative levels of the three types of protein?

A13: In the revised manuscript, we have analyzed *Lrig* mRNA and protein expression during adipogenic differentiation of MEFs. These analyses revealed that *Lrig1* was downregulated during adipocyte differentiation, *Lrig2* was transiently upregulated, whereas *Lrig3* levels appeared stable (Figure 1D-F and supplementary Figure S2A, D-F). The relative levels of the three *Lrig* proteins, i.e., *Lrig1* relative to *Lrig2* or *Lrig3* and *Lrig2* relative to *Lrig3*, could not be reliably determined because of technical challenges. That is, we do not have access to or are aware of any method that can quantify the endogenous levels of *Lrig1*, *Lrig2*, and *Lrig3* relative to each other. This is in contrast to the situation where we transduced or transfected cells with epitope-tagged LRIG variants, where we used the common epitope tag to compare the relative levels between the three LRIG proteins (e.g., Figure 4).

LRG null desensitized MEFs to response the BMPs, moreover there are deviation among bmp4, 6, and 9. It seems that it's not the receptors that lead to the difference as the authors have examined the level of different types of receptors. What's the possible target of LRIGs on BMP signaling? Do the LRIGs affect levels of BMPs? What's the possible reason for that maximal pSmad1/5 response did not appear to differ between the wild-type and Lrig-null MEFs (Fig.2b).

A14: In response to point 2 by reviewer #4, we have analyzed the expression of *Bmps*, *Bmprs*, *Ids*, and *Smads* in wild-type and *Lrig*-null MEFs (see A22). None of the analyzed genes showed altered expression that could explain the reduced BMP sensitivity among the *Lrig*-null cells. Regarding the possible molecular targets of LRIGs, we discuss that the type 1 receptors *BMPR1A* and *BMPR1B* but not *BMPR2* may be targets (Discussion section, page 15, lines 347-356). The similarity in the level of the maximally induced BMP response is intriguing, and we do not have a clear explanation for this observation at present. The detailed molecular mechanisms underlying the observed BMP sensitizing effects of LRIG proteins will be the subject of future studies.

Both smad and Mitogen-activated protein kinases/extracellular signal-regulated kinase (MAPK/ERK) pathway is supposed to be associated with the cell proliferation. Why the alterations of smad and MAPK in Lrig-null MEFs had no effect on cell numbers? To our knowledge, BMP4 can stimulate proliferation of C3H10t1/2, which was also derived from mouse embryonic fibroblasts. Regarding the Lrig-null change the sensitivity of cell responding to BMP4, would the author determined the proliferation rate after treating cells with different concentration of BMP4? Why Lrig-null MEFs in culture with about 5%FBS got higher OD value in MTT assay in Fig. S1d.

A15: The regulation of cell proliferation is certainly complex. Our MEFs were cultured in medium containing fetal bovine serum, which contains various growth factors, and potentially also growth inhibitors. For example, we know that the fetal bovine serum contains low levels of BMPs, as well as the MEF growth factor PDGF. Because of the biochemical complexity of animal sera, it is difficult to precisely predict the outcome of different serum-containing growth conditions. In the MTT assay, we observed a statistically significant difference ($p < 0.01$) between wild-type and *Lrig*-null cells in the presence of 5% FBS, but not 2.5% or 8% FBS. The observed difference at 5% FBS may certainly be real, although subtle, or it may represent statistical noise. In any case, we did not pursue this line of investigation.

Some alleles of Lrig1 associated with increased the BMI and hyperplasia of adipocytes? How do they affect LRIG1? Gain of function or loss of function?

A16: From the foregoing analyses, these alleles seem to confer the LRIG1 gene a gain of function. In the MEF experiments, *Lrig*-null MEFs showed deficient adipocyte differentiation compared with wild-type MEFs. With regards to BMP, *Lrig*-null MEFs had a lower BMP4 sensitivity than wild-type MEFs, and we further demonstrated that induction of LRIG1 or LRIG3 expression rescued the BMP sensitivity phenotype of *Lrig*-null MEFs. Additionally, *LRIG1* gene variants were associated with an increase in BMI in population-level studies, most likely through a mechanism of adipocyte hyperplasia, which is in our view, a gain of function.

Minor comment :

In Figure S1f,g, the authors examined the aerobic and anaerobic metabolism of WT and Lrig-null MEFs, what about OCR and ECAR in differentiated adipocytes derived those MEFs?

A17: This is an interesting question, but we have not analyzed the metabolism of differentiated adipocytes.

In figure 4, why the dox is 100ng/ml in A-D, while 500ng/ml in E-H?

A18: We apologize for the confusion regarding the doxycycline concentrations. There was no specific reason for us to use 100 ng/ml dox or 500 ng/ml dox. The doxycycline-inducible system used shows a maximal transcriptional response at doxycycline concentrations well below 100 ng/ml. This plateau in doxycycline-induced expression is apparent in different cellular systems, including in the MEFs used herein. Thus, there is no difference in the LRIG induction levels between 100 ng/ml and 500 ng/ml.

In figure 5A pSmad1/5 was analyzed following the cell were stimulated with 2.5ng/ml BMP4 for one hour, however in Figure 5b, c the concentration of BMP4 was 20ng/ml and treatment

lasted 20 minutes. Why the concentration and duration were different? If BMP4 in Figure 5b, c is the same as in Figure 5A, will it be different?

A19: The reasons for the different treatment regimens used in Figure 4A-C (Figure 5A-C in the original version) are as follows. Figure 4A and Figure 4B,C represent two different experimental designs. The purpose of the experiments shown in Figure 4 was to investigate the correlation between LRIG expression levels and the levels of BMP-induced pSmad1/5. In Figure 4A, we used virally transduced MEF lines with doxycycline-inducible LRIG alleles. Here, the majority of doxycycline treated cells induced the expression of the inducible gene (see, for example, Figure S2D). Accordingly, we used a standard stimulation protocol with 2.5 ng/ml BMP4 for one hour. In Figure 4B and C, in contrast, the cells were transiently transfected with the respective LRIG gene. In our hands, the MEFs are notoriously difficult to transfect, yielding only a low percentage of transformed cells. The low transformation efficiency represents a technical challenge; specifically, the specific pSmad1/5 signals from the small minority of transformed cells tend to drown in the noise from the background signals from the many non-transformed cells. To address this problem, we optimized the protocol to obtain an LRIG effect that was as strong as possible while keeping the background pSmad1/5 levels among the *Lrig*-null MEFs at a minimum. Thereby, we arrived at the protocol used in Figure 4B, C, which included the treatment of cells with 20 ng/ml BMP4 for 20 minutes.

In figure 7c, how was the Oil red O signal normalized? Is it divided by body length?

A20: All *C. elegans* genotypes studied were analyzed in parallel, and each experimental round was normalized to the combined mean signal, calculated from all genotype means. This information is stated in the Materials and Methods section (page 28, lines 683-685).

Reviewer #4 (Remarks to the Author):

The manuscript “LRIG proteins regulate lipid metabolism via BMP signaling and affect the risk of type 2 diabetes” by the group of Dr. Hedman provides evidence that mammalian LRIG proteins function as cellular BMP sensitizers and their ablation impairs adipogenesis in mouse embryonic fibroblasts. Lipid storage defects have also been shown in C. elegans with mutant LRIG/sma-10. More importantly, they showed a strong association between specific human LRIG1 gene variants with a decreased risk of type 2 diabetes, increased BMI, and altered adipocyte morphology, suggesting that LRIG proteins play important physiological roles in the regulation of lipid homeostasis in humans.

Overall this is a very well written and interesting study which is of high potential interest for scientists in this field, but a number of issues should be clarified to fully understand the observed findings.

1)Western blot analysis for LRIGs during different stages of adipogenesis in wild-type MEF lines would give an important information w.r.t adipocytic transformation.

A21: As described under A13, we performed qRT-PCR and Western blot analyses for the *Lrigs* during different stages of adipogenesis in wild-type MEFs. These results have been added to the revised manuscript (Figures 1D-F and S2E-J, respectively).

2) Did authors observe any difference in transcript levels of BMP4 (or other BMPs) and genes associated with BMP-signaling pathway (such as SMADs, ID genes etc) between the wild-type and *Lrig*-null MEF lines?

A22: We extended the gene expression analysis to include comparisons between Bmps, Smads, and Ids (Figure 2J). The results revealed no significant difference between the wild-type and *Lrig*-null MEFs with regard to their expression of Bmps, Bmprs, Smads, or Ids.

3) In Figure 2G, the blots for SMAD and P38 should be included (as they did for Figure 4E) and used for the respective calculations.

A23: Thank you for this comment. We have added the blots for Smad1 and p38 in Figure 2G.

Also, authors showed through western blot analysis of pSMAD1/5 (Figure 4E) that ectopic *LRIG1* rescued BMP signaling (upon stimulation with 20ng/ml BMP4), but no band was observed in *-dox* (empty vector/*LRIG*) or *+DOX* (empty vector) (or one can say in "*Lrig*-null MEFs") with 20 ng/ml BMP4 stimulation. Whereas a strong band for pSMAD1/5 was observed upon BMP4 stimulation (20ng/ml) in *Lrig*-null MEFs in Figure 2G. Please clarify.

A24: The apparent discrepancy between the results shown in Figure 4E and Figure 2G (in the previous version) may have different explanations. First, the visibility of the bands is dependent on the exposure times of the different blots. Thus, if the exposure time is increased for the blots in Figure 4E, F (previous version), pSmad bands could be readily observed for all cell lines and conditions after stimulation with 20 ng/ml BMP4. Thus, we have replaced the pSmad1/5 images in the former Figure 4E, F with blots with longer exposure times (Figure 3E, F in the revised manuscript). Second, a challenge of working with biological replicates is the considerable biological variation between different MEF clones. Thus, the blot shown in Figure 2G represents one experimental repeat with one biological replicate per genotype, and the blots in Figure 3E, F represent one experimental repeat of one single *LRIG1*-inducible MEF line (Figure 3E) or one single *LRIG3*-inducible MEF line (Figure 3F). Consequently, the results shown in Figure 2 and Figure 3 must be compared with caution because they were performed with different cell lines, and only one experimental repeat is shown. Importantly, however, the quantifications shown were based on four biological replicates and three experimental repeats (Figure 2H, I) or four experimental repeats with single cell lines (Figure 3G-L). We are confident that the experiments shown in Figures 2G-I and 3E-L show what they were designed to address, i.e., the experiment shown in Figure 2G-I shows that pp38 was slightly reduced in *Lrig*-null cells, and the experiment shown in Figure 3E-H demonstrates that *LRIG1*, but not *LRIG3*, could rescue this defect.

4) Authors are requested to show individual values (i.e. interleaved scatter plot) for the bar graphs in Figure 1 & 2.

A25: The bar graphs in Figure 1 and 2 have been replaced with interleaved scatter plots.

5) Using Immunofluorescence and western blot analysis of pSMAD/p38 (figure 4), authors demonstrate that ectopic *LRIG1* and *LRIG3* rescued BMP, but authors should also study the effects on lipid accumulation or markers of adipogenesis to see whether it restores adipogenesis in *Lrig*-null MEF lines or not?

A26: To address this issue, we have transduced an *Lrig*-null MEF line three times with either a dox-inducible LRIG1 allele, a dox-inducible LRIG3 allele, or a vector control. These cell lines were then used to investigate whether the restored LRIG expression could rescue the adipogenesis deficiency of *Lrig*-null MEFs. Clearly, dox-induced expression of LRIG1 or LRIG3, but not vector control, rescued the adipogenesis of *Lrig*-null MEFs. These new results have been included to the revised manuscript (page 9, lines 210-213 (Figure 3M)).

ADDITIONAL REVISIONS

A27: During our revision of the manuscript, we discovered an error in the normalization method for the Oil Red O experiments using *C. elegans*. When we initially ran the experiments, a non-BMP control genotype that was not included in the final figure was included in the normalization procedure. We have now corrected this normalization error and recalculated the statistics. Results from the Holm-Sidak's multiple comparisons test are presented below, and the figure (Figure 6C) has been modified accordingly. The normalization correction did not change any conclusions derived from the experiment.

PREVIOUS ERRONEOUS NORMALIZATION

	Mean Diff,	Significant?	Summary	Adjusted P Value
N2 vs. dpy-5	-0,4571	No	ns	0,1389
N2 vs. wk88	0,7553	Yes	*	0,0405
N2 vs. wk89	0,9408	Yes	*	0,0168
N2 vs. daf-4	0,7715	Yes	*	0,0405
N2 vs. sma-6	0,5067	No	ns	0,1389
N2 vs. sma-3	0,9111	Yes	*	0,0176

CORRECTED NORMALIZATION

	Mean Diff,	Significant?	Summary	Adjusted P Value
N2 vs. dpy-5	-0,4344	No	ns	0,1405
N2 vs. wk88	0,7253	Yes	*	0,0372
N2 vs. wk89	0,8979	Yes	*	0,0159
N2 vs. daf-4	0,741	Yes	*	0,0372
N2 vs. sma-6	0,4779	No	ns	0,1405
N2 vs. sma-3	0,8689	Yes	*	0,0167

A28: Due to over-crowding of Figure 2, we have moved the Western blots of BMP receptors (Figure 3B-D in the previous version) to the supplementary section of the revised manuscript (Figure S3D-F).

A29: Regarding the association analyses involving the GENiAL cohort, the GENiAL patient files have been updated which resulted in the exclusion of three individuals due to QC. This

exclusion resulted in minor changes of the numbers in Table S9 and the p-value on line 304 did change from $p=0.38$ to $p=0.39$. However, overall, the patient files update did not affect the results in any major way or any of the conclusions.

A30: The Materials and Methods section for the GENiAL study has been shortened because we can refer to a new publication describing the same genetic analyses (ref. 70 in the revised manuscript) (page 32, lines 771-772). The new reference also replaces Table S11 and references 70, 71, and 77.

References

O'Rourke, E.J., Soukas, A.A., Carr, C.E., and Ruvkun, G. (2009). *C. elegans* Major Fats Are Stored in Vesicles Distinct from Lysosome-Related Organelles. *Cell Metabolism* 10, 430–435.

Yu Y, Mutlu AS, Liu H, Wang MC. High-throughput screens using photo-highlighting discover BMP signaling in mitochondrial lipid oxidation. *Nat Commun.* 2017 Oct 11;8(1):865. doi: 10.1038/s41467-017-00944-3.PMID: 29021566

Reviewers' comments:

Reviewer #1 (Remarks to the Author):

The authors have satisfactorily addressed my concerns in this revision.

Reviewer #2 (Remarks to the Author):

The revised manuscript by Herdenberg et al. addressed most of my previous concerns. However, several issues persist.

The author attributed the low adipogenesis efficiency to the use of primary MEF lines. However, there are reports show high adipogenesis efficiency (more than 50%) even for the primary MEF (Duan Z, Zhao X, Fu X, et al. Tudor-SN, a novel coactivator of peroxisome proliferator-activated receptor γ protein, is essential for adipogenesis. *J Biol Chem.* 2014;289(12):8364-8374.; Cautivo KM, Lizama CO, Tapia PJ, et al. AGPAT2 is essential for postnatal development and maintenance of white and brown adipose tissue. *Mol Metab.* 2016;5(7):491-505.; Baudry A, Yang ZZ, Hemmings BA. PKB α is required for adipose differentiation of mouse embryonic fibroblasts. *J Cell Sci.* 2006;119(Pt 5):889-897.) .

If the Lrig function during differentiation, the downregulation of Lrig might promote adipogenesis as the author has shown Lrig1 was downregulated during adipocyte differentiation. In the study, Lrigs seem to function at pre-differentiation stage, because most data related to the effect of Lrigs on BMP signaling were got in the non-differentiated MEF cells. I am still curious about the result that Lrig-null MEFs in culture with about 5%FBS got higher OD value in MTT assay in Fig. S1d. Generally, if the cells survive and proliferate happily, they will got high differentiation rate after adipogenic induction, which is contradict to the present results. There should be some experiments and results to explain the confusion.

The relative expression levels of the three Lrig in cell models can be determined at mRNA level.

The author showed no altered expression of Bmpr1a and Bmpr1b as well as ALK-1 in Lrig-null cells, but still discussed that the type 1 receptors BMPR1A and BMPR1B but not BMPR2 may be targets for desensitizing BMP signaling (shown in text as "BMP4 and BMP6 specifically (interact with the type 1 receptors BMPR1A (ALK-3) and BMPR1B (ALK-6), whereas BMP9 specifically interacts with ACTRL1 (ALK-1)."). Is that reasonable?

In Figure 3A-C, pSmad1/5 was increased to very high level when concentration of BMP4 is 20ng/ml, and showed no difference between cells with and without Dox induction. So, it is supposed the non-transformed cells in Figure 4B-C will get the highly level of pSmad1/5 if the cells were treated with 20ng/ml BMP4. How does the author judge treatment of cells with 20ng/ml BMP4 for 20minutes will keep the background pSmad1/5 levels (Figure 4B-C) among the Lrigs-null MEFs at a minimum?

Reviewer #3 (Remarks to the Author):

The authors responded to all my comments and added new experiments and data. I very much appreciate their additional data.

Rebuttal letter

Dear Editors and Referees,

Thank you for critically reviewing our revised manuscript titled, "LRIG proteins regulate lipid metabolism via BMP signaling and affect the risk of type 2 diabetes". We are delighted to note that reviewers #1 and #3 were satisfied with our revisions, and we also appreciate the additional questions raised by reviewer #2. Attached is our further revised manuscript. All changes made to the manuscript can be viewed in the tracked changes in the attached Word file. A point-by-point rebuttal to all of the reviewers' comments is provided below, with the comments displayed in italics and our responses (A1-A7) in standard font. Line numbers mentioned in our responses refer to the line numbering of the manuscript in the Article file.

We hope that you will find our revisions adequate and the revised manuscript suitable for publication in Communications Biology.

Sincerely,
Håkan Hedman, corresponding author

Reviewer #1 (Remarks to the Author):

The authors have satisfactorily addressed my concerns in this revision.

A1: We are glad that reviewer #1 was satisfied with our revision, and we thank them for investing their time and effort in improving our manuscript.

Reviewer #2 (Remarks to the Author):

The revised manuscript by Herdenberg et al. addressed most of my previous concerns. However, several issues persist.

The author attributed the low adipogenesis efficiency to the use of primary MEF lines. However, there are reports show high adipogenesis efficiency (more than 50%) even for the primary MEF (Duan Z, Zhao X, Fu X, et al. Tudor-SN, a novel coactivator of peroxisome proliferator-activated receptor γ protein, is essential for adipogenesis. J Biol Chem. 2014;289(12):8364-8374.; Cautivo KM, Lizama CO, Tapia PJ, et al. AGPAT2 is essential for postnatal development and maintenance of white and brown adipose tissue. Mol Metab. 2016;5(7):491-505.; Baudry A, Yang ZZ, Hemmings BA. PKBalpha is required for adipose differentiation of mouse embryonic fibroblasts. J Cell Sci. 2006;119(Pt 5):889-897.) .

A2: We appreciate that primary MEF lines may show highly efficient adipogenesis. However, the adipogenesis efficiency varies widely among different MEF lines. For example, the commonly used 3T3-L1 line shows a very high efficiency [e.g., Duan et al., 2014], whereas the Flp-In-3T3 line does not appear to be able to differentiate into adipocytes, at all [Dastagir et al., 2014]. Our MEF lines were established using the 3T3 protocol, and their adipogenic potentials seem to fall between the extremes of these two 3T3 MEF lines. It could also be relevant to note that the differentiation potential of MEFs seems to decrease as the MEFs are

increasingly transformed. In this regard, our MEFs had undergone spontaneous immortalization according to the 3T3 protocol, been genetically modified using adenoviral and retroviral transductions, and been subjected to selection procedures including different antibiotics and fluorescence-activated cell sorting. Thus, they had undergone both uncontrolled and controlled genetic changes and circumstantially been selected for rapid proliferation via extensive cultivation. Given this background of our MEF lines, we do not find their relatively low adipogenesis efficiency to be extraordinary. Therefore, and for the sake of not extending the Discussion unnecessarily, we would prefer not to further comment on this in the manuscript.

If the Lrig function during differentiation, the downregulation of Lrig might promote adipogenesis as the author has shown Lrig1 was downregulated during adipocyte differentiation. In the study, Lrigs seem to function at pre-differentiation stage, because most data related to the effect of Lrigs on BMP signaling were got in the non-differentiated MEF cells. I am still curious about the result that Lrig-null MEFs in culture with about 5%FBS got higher OD value in MTT assay in Fig. S1d. Generally, if the cells survive and proliferate happily, they will get high differentiation rate after adipogenic induction, which is contradict to the present results. There should be some experiments and results to explain the confusion.

A3: This point by reviewer #2 is well taken; however, it should also be noted that the role of cell proliferation among different cell populations during adipogenesis remains controversial [Marquez et al., 2017]. Regarding our results, and as discussed in our previous rebuttal (A15), the observed difference in the MTT assay with 5% FBS may be real, although subtle, or it may represent statistical noise. Nevertheless, we did not observe any difference in cell proliferation rates at higher or lower FBS concentrations. Importantly, we performed all of our adipogenesis experiments with 10% FBS, a concentration that did not result in any difference in cell proliferation. Nevertheless, to acknowledge the observed difference in the apparent proliferation rates with 5% FBS, we added a sentence to note this observation in the Results section, pages 5-6, lines 114-116.

The relative expression levels of the three Lrig in cell models can be determined at mRNA level.

A4: The relative mRNA expression levels of *Lrig1*, *Lrig2*, and *Lrig3* during MEF adipogenesis *in vitro* are shown in Figure 1D-F. That is, we determined the relative expression levels of each *Lrig* mRNA individually. However, the apparent levels of the three *Lrig* mRNAs cannot be directly compared, i.e., *Lrig1* cannot be compared with *Lrig2* or *Lrig3*, and *Lrig2* cannot be compared with *Lrig3* because we do not know the efficiency of the respective qRT-PCR assay.

The author showed no altered expression of Bmpr1a and Bmpr1b as well as ALK-1 in Lrig-null cells, but still discussed that the type 1 receptors BMPR1A and BMPR1B but not BMPR2 may be targets for desensitizing BMP signaling (shown in text as “BMP4 and BMP6 specifically (interact with the type 1 receptors BMPR1A (ALK-3) and BMPR1B (ALK-6), whereas BMP9 specifically interacts with ACTRL1 (ALK-1).”). Is that reasonable?

A5: Yes, we believe that BMPR1A (ALK-3) and BMPR1B (ALK-6) may be the targets of the LRIG proteins despite the observation that their expression levels appear unaltered. Possible regulatory mechanisms that do not include altered protein expression levels include various posttranscriptional modifications and altered subcellular localization. The latter mechanism

has some experimental support in *C. elegans* and is briefly discussed in the Discussion, pages 15-16, lines 360-363.

In Figure 3A-C, pSmad1/5 was increased to very high level when concentration of BMP4 is 20ng/ml, and showed no difference between cells with and without Dox induction. So, it is supposed the non-transformed cells in Figure 4B-C will get the highly level of pSmad1/5 if the cells were treated with 20ng/ml BMP4. How does the author judge treatment of cells with 20ng/ml BMP4 for 20minutes will keep the background pSmad1/5 levels (Figure 4B-C) among the *Lrig*-null MEFs at a minimum?

A6: Indeed, after 1 hour of BMP4 stimulation at 20 ng/ml, there was no difference in pSmad1/5 levels between cells with and without doxycycline-induced LRIG1/3 expression (Figure 3A-C). However, at earlier time points there was a clear difference between *Lrig*-expressing (wild-type, WT) and *Lrig*-null (TKO) MEFs that were stimulated with 20 ng/ml BMP4, as shown in the kinetic experiment included below. In the experiments shown in Figure 4B and C, we faced a technical problem; i.e., because of the poor transfection efficiencies of the MEFs, the specific pSmad1/5 signals from the small minority of transformed *LRIG*-expressing cells tended to be lost in the noise from the background signals from the many non-transformed *Lrig*-null cells. To address this signal-to-noise problem, we optimized the protocol to obtain a measurable LRIG effect while keeping the background pSmad1/5 levels among the *Lrig*-null MEFs at a minimum. Thus, the treatment of cells with 20 ng/ml BMP4 for 20 minutes (as in Figure 4B, C) yielded an LRIG effect that was readily measurable while keeping the background pSmad1/5 levels among the *Lrig*-null MEFs close to zero. Importantly, the proper controls were included to ascertain that the observed effects were caused by the respective LRIG1 variants (Figure 4B, C). For clarity, the rationale for using the modified BMP4 stimulation protocol for the transiently transfected cells has been added to the Results section, page 10, lines 230-235.

Reviewer #3 (Remarks to the Author):

The authors responded to all my comments and added new experiments and data. I very much appreciate their additional data.

A7: We are glad that reviewer #3 was satisfied with our revision, and we thank them for investing their time and effort in improving our manuscript.

References

Dastagir K, Reimers K, Lazaridis A, Jahn S, Maurer V, Strauß S, Dastagir N, Radtke C, Kampmann A, Bucan V, Vogt PM. Murine embryonic fibroblast cell lines differentiate into three mesenchymal lineages to different extents: new models to investigate differentiation processes. *Cell Reprogram.* 2014 Aug;16(4):241-52. doi: 10.1089/cell.2014.0005.PMID: 25068630

Duan Z, Zhao X, Fu X, Su C, Xin L, Saarikettu J, Yang X, Yao Z, Silvennoinen O, Wei M, Yang J. Tudor-SN, a novel coactivator of peroxisome proliferator-activated receptor \$\gamma\$ protein, is essential for adipogenesis. *J Biol Chem.* 2014 Mar 21;289(12):8364-74. doi: 10.1074/jbc.M113.523456. Epub 2014 Feb 12.PMID: 24523408

Marquez MP, Alencastro F, Madrigal A, Jimenez JL, Blanco G, Gureghian A, Keagy L, Lee C, Liu R, Tan L, Deignan K, Armstrong B, Zhao Y. The Role of Cellular Proliferation in Adipogenic Differentiation of Human Adipose Tissue-Derived Mesenchymal Stem Cells. *Stem Cells Dev.* 2017 Nov 1;26(21):1578-1595. doi: 10.1089/scd.2017.0071. Epub 2017 Oct 4.PMID: 28874101

REVIEWERS' COMMENTS:

Reviewer #2 (Remarks to the Author):

The authors have addressed my concerns in this revision.